# Otoferlin acts as a Ca²⁺ sensor for vesicle fusion and vesicle pool replenishment at auditory hair cell ribbon synapses

Nicolas Michalski[1,2,3]*, Juan D Goutman[4], Sarah Marie Auclair[5],
Jacques Boutet de Monvel[1,2,3], Margot Tertrais[2,6], Alice Emptoz[1,2,3],
Alexandre Parrin[1,2,3], Sylvie Nouaille[1,2,3], Marc Guillon[7], Martin Sachse[8],
Danica Ciric[1,2,3], Amel Bahloul[1,2,3,9], Jean-Pierre Hardelin[1,2,3],
Roger Bryan Sutton[10,11], Paul Avan[12,13,14], Shyam S Krishnakumar[5,15],
James E Rothman[5,15], Didier Dulon[2,6†], Saaid Safieddine[1,2,3,9†],
Christine Petit[1,2,3,16,17†]*

[1]Unité de Génétique et Physiologie de l'Audition, Institut Pasteur, Paris, France;
[2]UMRS 1120, Institut National de la Santé et de la Recherche Médicale, Paris,
France; [3]Sorbonne Universités, UPMC Université Paris 06, Complexité du Vivant,
Paris, France; [4]Instituto de Investigaciones en Ingeniería Genética y Biología
Molecular, Consejo Nacional de Investigaciones Científicas y Técnicas, Buenos
Aires, Argentina; [5]Department of Cell Biology, Yale University School of Medicine,
New Haven, United States; [6]Laboratoire de Neurophysiologie de la Synapse
Auditive, Bordeaux Neurocampus, Université de Bordeaux, Bordeaux, France;
[7]Wave Front Engineering Microscopy Group, Neurophotonics Laboratory, Centre
National de la Recherche Scientifique, UMR 8250, University Paris Descartes,
Sorbonne Paris Cité, Paris, France; [8]Center for Innovation & Technological
Research, Ultrapole, Institut Pasteur, Paris, France; [9]Centre National de la
Recherche Scientifique, France; [10]Department of Cell Physiology and Molecular
Biophysics, Texas Tech University Health Sciences Center, Lubbock, United States;
[11]Center for Membrane Protein Research, Texas Tech University Health Sciences
Center, Lubbock, United States; [12]Laboratoire de Biophysique Sensorielle,
Université Clermont Auvergne, Clermont-Ferrand, France; [13]UMR 1107, Institut
National de la Santé et de la Recherche Médicale, Clermont-Ferrand, France;
[14]Centre Jean Perrin, Clermont-Ferrand, France; [15]Department of Clinical and
Experimental Epilepsy, Institute of Neurology, University College London, London,
United Kingdom; [16]Syndrome de Usher et Autres Atteintes Rétino-Cochléaires,
Institut de la Vision, Paris, France; [17]Collège de France, Paris, France

*For correspondence:
nicolas.michalski@pasteur.fr (NM);
christine.petit@pasteur.fr (CP)

†These authors contributed
equally to this work

Competing interest: See
page 30

Reviewing editor: Christian
Rosenmund, Charité-
Universitätsmedizin Berlin,
Germany

**Abstract** Hearing relies on rapid, temporally precise, and sustained neurotransmitter release at
the ribbon synapses of sensory cells, the inner hair cells (IHCs). This process requires otoferlin, a six
C₂-domain, Ca²⁺-binding transmembrane protein of synaptic vesicles. To decipher the role of
otoferlin in the synaptic vesicle cycle, we produced knock-in mice (Otof $^{Ala515,Ala517/Ala515,Ala517}$) with
lower Ca²⁺-binding affinity of the C₂C domain. The IHC ribbon synapse structure, synaptic Ca²⁺
currents, and otoferlin distribution were unaffected in these mutant mice, but auditory brainstem
response wave-I amplitude was reduced. Lower Ca²⁺ sensitivity and delay of the fast and sustained
components of synaptic exocytosis were revealed by membrane capacitance measurement upon
modulations of intracellular Ca²⁺ concentration, by varying Ca²⁺ influx through voltage-gated Ca²⁺-

channels or $Ca^{2+}$ uncaging. Otoferlin thus functions as a $Ca^{2+}$ sensor, setting the rates of primed vesicle fusion with the presynaptic plasma membrane and synaptic vesicle pool replenishment in the IHC active zone.

DOI: https://doi.org/10.7554/eLife.31013.001

## Introduction

The extremely precise encoding of sound temporal features by the first synapse of the mammalian auditory system, that is, between the sensory inner hair cell (IHC) and the primary auditory neuron, is crucial for many perceptive tasks. It is involved in periodicity-pitch detection, prosody cue detection, and sound source localization required for voice and melody identification, speech perception, and auditory scene analysis, respectively (*Schnupp et al., 2011*). Sound-evoked mechanical stimulation of the IHC sensory antenna, the hair bundle, induces changes in membrane potential, modulating synaptic exocytosis with submillisecond precision (*Glowatzki and Fuchs, 2002*; *Goutman, 2012*; *Li et al., 2014*). This temporal precision exceeds that for most conventional synapses, and allows sound-evoked action potentials of the primary auditory neurons to be phase-locked to the sinusoidal acoustic signal up to frequencies of ~4 kHz (*Fuchs, 2005*; *Moser et al., 2006*; *Safieddine et al., 2012*). In addition, IHCs can maintain neurotransmitter release at high frequency for several minutes (*Kiang, 1965*), which implies a continuous supply of an unusually large number of vesicles to the synaptic active zones. The basolateral region of IHCs contains 10 to 30 synaptic active zones, each of which faces the single dendritic bouton of a primary auditory neuron. Most of the vesicles in each synapse are tethered to a ribbon-shaped osmiophilic structure (hence the name 'ribbon synapse'), presumably forming a pool of primed vesicles for the immediate and sustained replenishment of the pool of fusion-competent vesicles located between the base of the ribbon and the presynaptic plasma membrane (*von Gersdorff and Matthews, 1997*; *Lenzi et al., 1999*; *Moser and Beutner, 2000*).

Mature IHCs lack several common synaptic proteins (*Safieddine and Wenthold, 1999*; *Vogl et al., 2015*). The molecular composition of the exocytosis machinery underlying the functional features of IHC synapses remains largely unknown. In particular, mature IHC synapses lack the synaptic vesicle transmembrane proteins synaptotagmin 1 and 2 (Syt1 and Syt2) (*Safieddine and Wenthold, 1999*; *Beurg et al., 2010*), which function as $Ca^{2+}$ sensors for rapid, synchronous neurotransmitter release at central nervous system synapses (*Südhof, 2013*). These proteins, which contain two cytoplasmic $C_2$-domains, bind to membrane phospholipids in a $Ca^{2+}$-dependent manner (*Brose et al., 1992*; *Sutton et al., 1995*; *Wang et al., 2014*), and trigger the final steps of synaptic exocytosis by interacting with complexin and the SNARE molecular complex (*Bennett et al., 1992*; *Söllner et al., 1993*; *Li et al., 1995*; *Giraudo et al., 2006*; *Südhof, 2013*). However, unlike vesicles of central nervous system synapses, IHC vesicles contain otoferlin, a $Ca^{2+}$-binding single-pass membrane protein with six $C_2$ domains ($C_2$A-F) and two Fer domains in its cytoplasmic region (*Yasunaga et al., 1999*; *Roux et al., 2006*; *Lek et al., 2010*). Otoferlin belongs to the ferlin family, which is thought to have originated earlier than synaptotagmins and E-synaptotagmins during evolution (*Lek et al., 2012*) (*Figure 1A*). Otoferlin, defective in a recessive form of profound congenital deafness (*Yasunaga et al., 1999*; *Roux et al., 2006*; *Lek et al., 2010*), is required for normal synaptic exocytosis in auditory (*Roux et al., 2006*) and vestibular hair cells (*Dulon et al., 2009*). Its role in the hair cell synaptic vesicle cycle remains unclear. IHC synaptic exocytosis is almost entirely abolished in adult mutant mice lacking otoferlin (*Otof$^{-/-}$* mice), despite normal $Ca^{2+}$ currents and ribbon synapse morphogenesis (*Roux et al., 2006*). This finding, together with the absence of Syt1, Syt2, and Syt9 from mature IHCs (*Safieddine and Wenthold, 1999*; *Beurg et al., 2010*), has led to the hypothesis that otoferlin acts as the major $Ca^{2+}$ sensor triggering synaptic vesicle fusion with the plasma membrane in the IHC active zone (*Roux et al., 2006*). However, the IHC synapses of *Otof$^{-/-}$* mice being silent, these mice cannot be used to determine in which step(s) of the synaptic vesicle cycle otoferlin exerts its putative $Ca^{2+}$ sensing role. The *Pachanga* mutant, a deaf mouse harboring a missense mutation in the otoferlin $C_2$F domain, showed unaffected vesicle fusion but a major decrease of the sustained component of IHC synaptic exocytosis (*Pangrsic et al., 2010*), which led Pangrsic and coll. to suggest a role for otoferlin in synaptic vesicle pool replenishment. However, the $Ca^{2+}$ sensing role of otoferlin could not be assessed in the *Pachanga* mice because the mutation

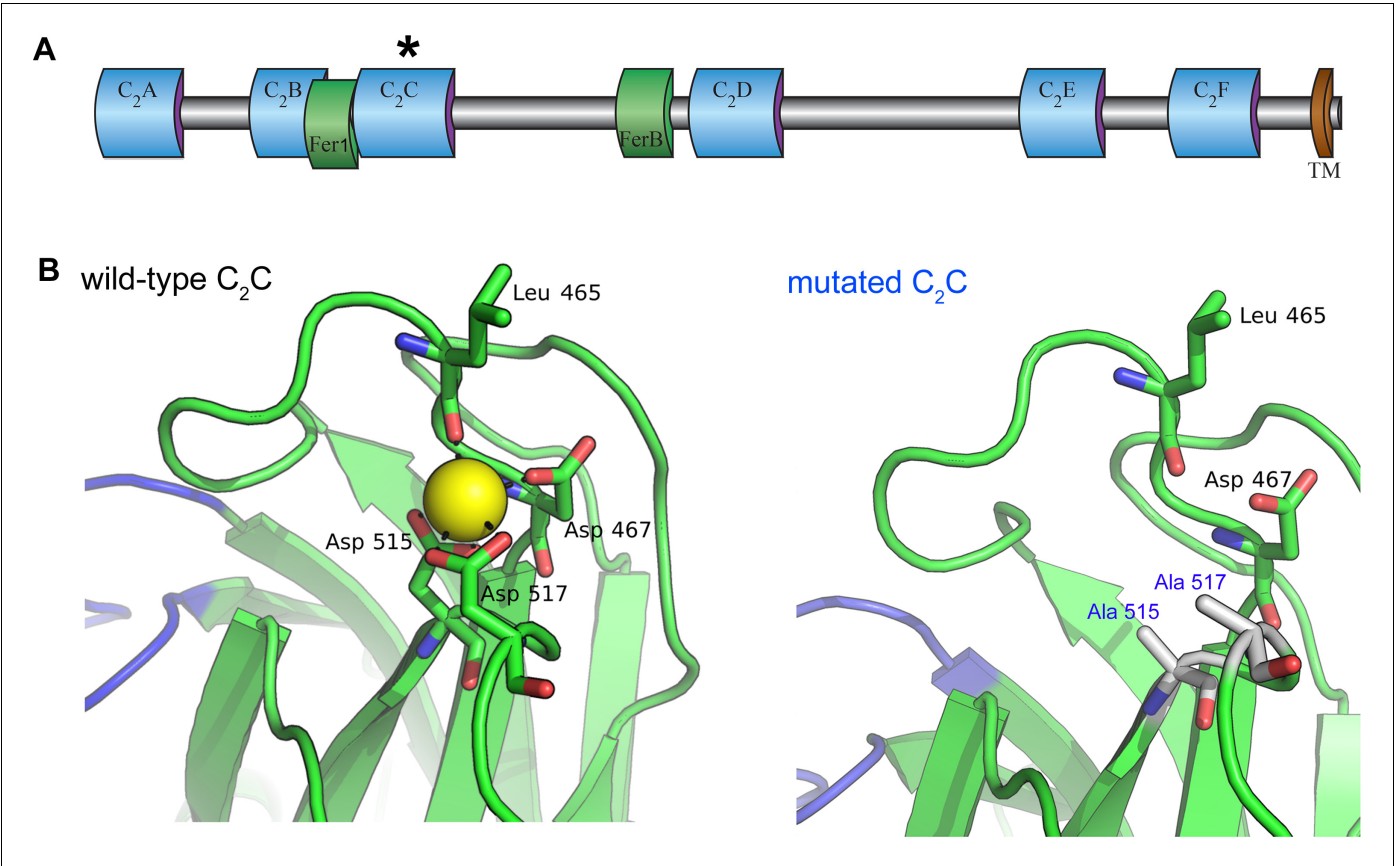

**Figure 1.** Homology model of the otoferlin $C_2C$ domain. (**A**) Predicted domain structure of the otoferlin protein. Otoferlin is a transmembrane vesicular protein (1997 amino acids in the mouse) consisting of six $C_2$ domains ($C_2$A-F), one Fer1 domain, and one FerB domain (central domains of still unknown function in proteins of the ferlin family) in its extravesicular (i.e cytoplasmic) portion. The asterisk indicates the $C_2C$ domain targeted for mutagenesis. (**B**) Ten candidate models were calculated with MODELLER (**Webb and Sali, 2014**), based on the X-ray structure of the human dysferlin $C_2A$ domain (4ihb) (**Fuson et al., 2014**). The primary sequence identity between dysferlin $C_2A$ and mouse otoferlin $C_2C$, assuming type II $C_2$ domain topology, is 23%. The model with the lowest energy score was selected for analysis. Further refinement was performed with ModRefiner (**Xu and Zhang, 2011**) using dysferlin $C_2A$ as the structural reference. At least one $Ca^{2+}$ ion (yellow sphere) could be modeled in the $C_2C$ domain of otoferlin. The position and coordination of this ion within the putative divalent cation binding pocket of the homology model was refined by simulating 20 ns of molecular dynamics using NAMD (**Phillips et al., 2005**).

DOI: https://doi.org/10.7554/eLife.31013.002

does not target a $Ca^{2+}$-binding site and the otoferlin cell content is much lower than normal. Nevertheless, the results obtained in these mice raise the possibility that a deficiency in the dynamics of vesicle pool replenishment explains the synaptic exocytosis defect in $Otof^{-/-}$ mice, despite their unaffected number of synaptic vesicles at the presynaptic zone (**Roux et al., 2006**). Of note, $Otof^{I515T/I515T}$ knock-in mice, harboring a Ile515Thr missense mutation in the $C_2C$ domain, also display abnormal synaptic exocytosis and sustained release (**Strenzke et al., 2016**). Similarly to the *Pachanga* mutation, the Ile515Thr mutation does not affect the $Ca^{2+}$-binding site of the $C_2$ domain, and results in a strongly reduced otoferlin content, again precluding any conclusion regarding a possible $Ca^{2+}$ sensing role of otoferlin in synaptic vesicle pool replenishment. Finally, a possible role of otoferlin in synaptic endocytosis and the reformation of correctly sized vesicles has been suggested, based on the in vitro interaction between otoferlin and the AP-2 adaptor protein complex involved in clathrin-mediated endocytosis (**Duncker et al., 2013**), the presence of large abnormal endosome-like vacuoles containing otoferlin in the IHCs of mutant mice lacking AP-2μ (**Revelo et al., 2014**; **Jung et al., 2015**), and the presence of enlarged otoferlin-immunoreactive vesicular structures, potentially of endosomal origin, in $Otof^{I515T/I515T}$ mutant mice (**Strenzke et al., 2016**).

We sought to identify the step(s) of the synaptic vesicle cycle at which otoferlin may act as a $Ca^{2+}$ sensor, by generating a mutant mouse line carrying otoferlin mutations modifying the binding of $Ca^{2+}$ ions to a $C_2$ domain whilst preserving both protein levels and the ultrastructure of the IHC ribbon synapse. By studying these mice, we were able to demonstrate the $Ca^{2+}$ sensing role of otoferlin both in the fusion of synaptic vesicles with the presynaptic membrane and in vesicle pool replenishment.

## Results

### Genetic modification of the otoferlin $C_2C$ domain $Ca^{2+}$-binding site

We investigated the roles of otoferlin in the IHC synaptic vesicle cycle through a mutagenesis strategy similar to that previously used to demonstrate that Syt1 and Syt2 function as $Ca^{2+}$ sensors for fast exocytosis, and that Syt7 functions as the $Ca^{2+}$ sensor for synaptic facilitation, at central nervous system synapses (*Fernández-Chacón et al., 2001*; *Schneggenburger et al., 2012*; *Jackman et al., 2016*). The $Ca^{2+}$-binding pockets of the Syt $C_2$-domains consist of a cluster of conserved aspartic acid residues surrounded by a ring of positively charged residues (*Sutton et al., 1995*; *Shao et al., 1996*). Substitution of any of these residues reduces the $Ca^{2+}$-binding affinity of Syt1 and Syt2, decreasing the $Ca^{2+}$ sensitivity of exocytosis, or that of Syt7, eliminating facilitation (*Fernández-Chacón et al., 2001*; *Schneggenburger et al., 2012*; *Jackman et al., 2016*). The otoferlin $C_2A$ domain does not bind $Ca^{2+}$, but the $C_2$ B-F domains have sizeable in vitro $Ca^{2+}$-binding affinities (about 13–25 µM) (*Johnson and Chapman, 2010*). The $C_2C$ and $C_2F$ domains have been shown to interact specifically with phosphatidylinositol 4,5-bisphosphate [PI(4,5)P2] in vitro, suggesting a possible role in mediating the preferential membrane targeting of otoferlin (*Padmanarayana et al., 2014*). In addition, the neutralization of two aspartic acid residues (Asp515 and Asp517) in the otoferlin $C_2C$ domain has been shown to prevent $C_2C$ domain-mediated membrane fusion in in vitro assays (*Johnson and Chapman, 2010*). These results prompted us to target the $C_2C$ domain for in vivo mutagenesis. Using the crystal structure of the dysferlin $C_2A$ domain as a template (*Fuson et al., 2014*), we constructed a homology model of otoferlin $C_2C$, to predict the structure of the $Ca^{2+}$-binding site of the $C_2C$ domain. Otoferlin $C_2C$ probably folds into a typical type-II $C_2$ domain. At least one $Ca^{2+}$ ion could be bound by three aspartic acid residues (Asp467, Asp515, and Asp517) located on top loops 1 and 3 of the $C_2C$ domain. The substitution of two of these aspartic acid residues by alanine residues (Asp515Ala and Asp517Ala) is predicted to affect $Ca^{2+}$ binding (*Figure 1B*). We therefore generated a knock-in mouse line carrying these two missense mutations in the homozygous state, *Otof* [Ala515,Ala517/Ala515,Ala517] mice (hereafter referred to as *Otof* [C2C/C2C] mice), by homologous recombination (see Materials and methods).

### *Otof* [C2C/C2C] mice have abnormal auditory nerve fiber responses

We first recorded auditory brainstem responses (ABRs), to monitor the electrical response of the primary auditory neurons and the successive neuronal relays of the central auditory pathway to brief sound stimuli, in *Otof* [+/+], *Otof* [C2C/+], and *Otof* [C2C/C2C] mice. At the age of one month, ABR thresholds were similar in *Otof* [+/+] (n = 5) and *Otof* [C2C/+] (n = 4) mice (*Figure 2—figure supplement 1A*, p>0.3), but they were slightly higher in *Otof* [C2C/C2C] mice (n = 11), by about 5.4 ± 3.3 dB, on average, than in *Otof* [C2C/+] mice (n = 9), for all frequencies tested (*Figure 2A*; two-way-ANOVA, p=0.04). Strikingly, the mean peak amplitude for ABR wave-I (in response to 100–200 tone bursts), reflecting the synchronous electrical response of primary afferent neurons, was lower in *Otof* [C2C/C2C] mice (n = 7) than in *Otof* [C2C/+] mice (n = 7), by a factor of 2.7 (*Figure 2B–C and E*) (p=0.006). This lower ABR wave-I amplitude did not affect ABR waves-II to V, which reflect the mean neuronal activity of the cochlear nucleus and higher auditory relays (*Figure 2B*). Earlier in development, on postnatal days 16–21 (P16-P21), the ABR thresholds of *Otof* [C2C/C2C] mice (n = 7) were similar to those of *Otof* [C2C/+] mice (n = 8; *Figure 2D*; two-way-ANOVA, p>0.2), but the wave-I amplitude was already lower for all sound intensities tested between 35 dB and 105 dB (*Figure 2D*; two-way-ANOVA, p<$10^{-4}$). From the age of three months onwards, ABR thresholds increased in *Otof* [C2C/C2C] mice (n = 5–6) to reach, on average, values 19 ± 5.8 dB higher than those in *Otof* [C2C/+] mice (n = 5; *Figure 2—figure supplement 1B*; two-way-ANOVA, p<$10^{-4}$ for the 3-month-old and 5-month-old mouse groups). The peak amplitude of the ABR wave-I also decreased further by a factor

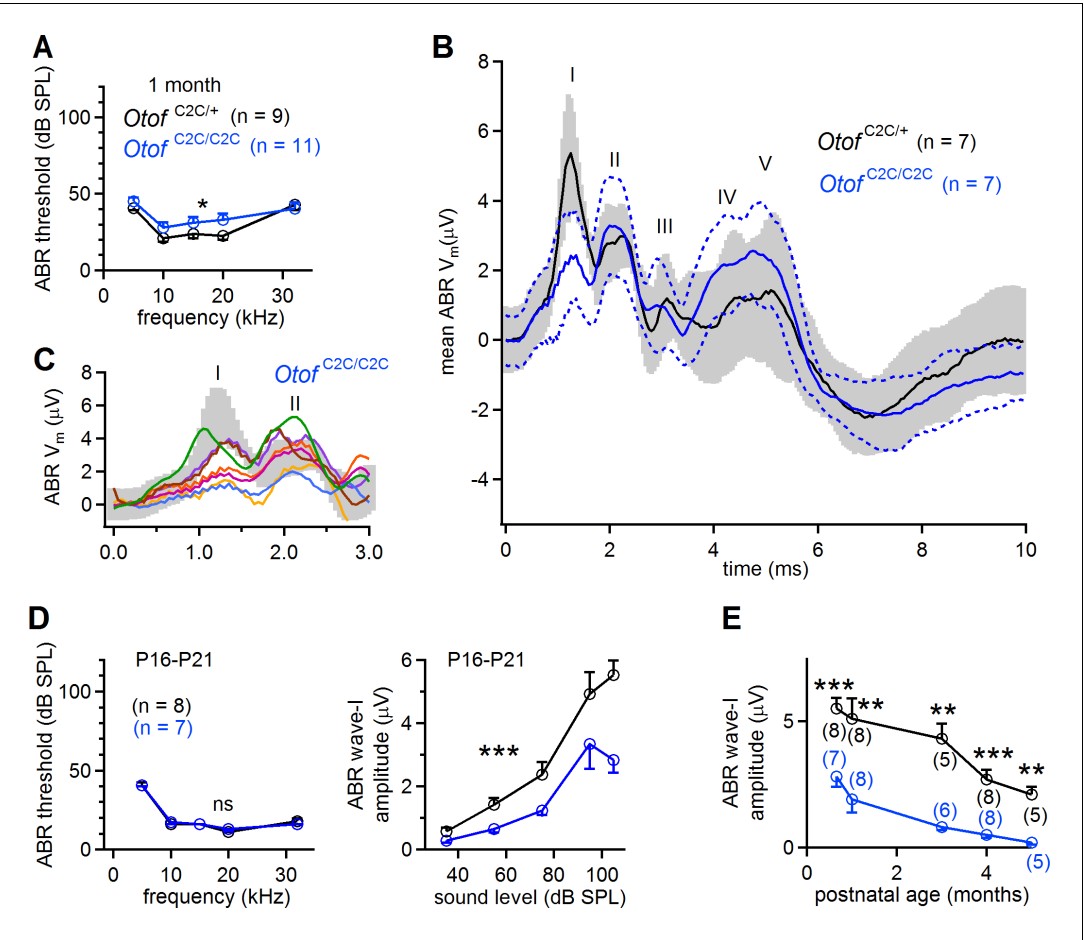

**Figure 2.** Smaller amplitude of ABR wave-I in *Otof*<sup>C2C/C2C</sup> mice. (**A**) ABR thresholds of one-month-old *Otof*<sup>C2C/+</sup> (black) and *Otof*<sup>C2C/C2C</sup> (blue) mice, for pure tone frequencies between 5 kHz and 32 kHz. (**B**) Mean ABR trace recorded in *Otof*<sup>C2C/+</sup> (black line, n = 7 mice) and *Otof*<sup>C2C/C2C</sup> (blue line, n = 7 mice) mice upon a 95 dB SPL (sound pressure level) pure tone stimulation at 10 kHz. The gray area and the area between the blue dotted lines denote the overlapping 95% confidence intervals of the traces recorded in *Otof*<sup>C2C/+</sup> and *Otof*<sup>C2C/C2C</sup> mice, respectively. (**C**) Detailed view of ABR wave-I from individual ABR recordings in the *Otof*<sup>C2C/C2C</sup> mice, used to calculate the mean traces in (**B**). Each colored trace corresponds to an individual *Otof*<sup>C2C/C2C</sup> mouse. The 95% confidence intervals of the mean traces in (**B**) are shown in gray for *Otof*<sup>C2C/+</sup> mice. (**D**) *Left*: ABR thresholds of P16-P21 *Otof*<sup>C2C/+</sup> (black) and *Otof*<sup>C2C/C2C</sup> (blue) mice, for pure tone frequencies between 5 kHz and 32 kHz. *Right*: Plot of the amplitude of ABR wave-I as a function of sound level for a 10 kHz pure tone. (**E**) Plot of ABR wave-I amplitude against age (sound at 10 kHz and 105 dB SPL; the numbers indicated in parentheses correspond to the number of mice studied at each age). Data information: In (**A, D**), data are presented as the mean ±SEM. *p<0.05, ***p<0.001, ns not significant (two-way-ANOVA test). In (**B**), data are presented as the mean and its 95% confidence intervals. In (**E**), data are presented as the mean ± SEM. **p<0.01, ***p<0.001 (Student's *t*-test with Welch correction).

DOI: https://doi.org/10.7554/eLife.31013.003

The following figure supplement is available for figure 2:

**Figure supplement 1.** Progressive hearing loss in *Otof*<sup>C2C/C2C</sup> mice after one month of age.

DOI: https://doi.org/10.7554/eLife.31013.004

of about 10 (p<0.01, for all ages, with a minimum of 5 animals per group; *Figure 2E*). By contrast, distortion-product otoacoustic emissions (DPOAEs), which probe outer hair cell (OHC) function, were similar in *Otof*<sup>C2C/+</sup> (n = 5) and *Otof*<sup>C2C/C2C</sup> (n = 5–8) mice, in terms of both threshold and amplitude. The cochlear amplification of sound stimuli was, thus, preserved in homozygous mutant mice (*Figure 2—figure supplement 1C*; two-way-ANOVA, p=0.6 and p=0.1 for the 1-month-old and 5-month-old mouse groups, respectively). Overall, the auditory phenotype of *Otof*<sup>C2C/C2C</sup> mice

is consistent with the restriction of otoferlin dysfunction to IHCs (*Roux et al., 2006*), with the decrease in ABR wave-I amplitude suggesting a dysfunction of the ribbon synapses.

## *Otof* $^{C2C/C2C}$ IHCs have normal otoferlin contents and ribbon synapse ultrastructure

We analyzed the IHC ribbon synapses, immunofluorescently labeled for ribeye (a core ribbon protein), the presynaptic L-type Ca$^{2+}$ channel Ca$_v$1.3, and the postsynaptic glutamate receptor GluA2 (*Figure 3—figure supplement 1*), by confocal microscopy, in *Otof* $^{C2C/+}$ and *Otof* $^{C2C/C2C}$ mice on P15-P17 (i.e., a few days after hearing onset). All ribeye-immunoreactive ribbons in *Otof* $^{C2C/+}$ and *Otof* $^{C2C/C2C}$ IHCs displayed Ca$_v$1.3 staining systematically apposed to the GluA2 subunit staining. The mean number of ribbons per IHC did not differ significantly between *Otof* $^{C2C/+}$ mice (16.4 ± 0.2) and *Otof* $^{C2C/C2C}$ mice (16.2 ± 0.2) (63 IHCs from the apical coil of four mice per genotype, p=0.4). In mature IHCs of *Otof* $^{C2C/+}$ and *Otof* $^{C2C/C2C}$ mice, otoferlin was detected throughout the cytosol, with intense immunolabeling of the basolateral region containing the ribbon synapses, whereas it was undetectable in the IHCs of *Otof* $^{-/-}$ mice, as previously reported (*Roux et al., 2006*) (*Figure 3A–B*). The immunofluorescence levels of otoferlin at the apex, middle, and base of IHCs were similar between *Otof* $^{C2C/+}$ and *Otof* $^{C2C/C2C}$ mice (p>0.3 for all comparisons; *Figure 3C*), indicating that levels of otoferlin expression were unaffected in *Otof* $^{C2C/C2C}$ IHCs (*n* = 51 cells from seven mice for each genotype).

We also compared the distributions of immunostaining for otoferlin and the vesicular glutamate transporter Vglut3, imaged at high resolution by stimulated emission depletion microscopy (STED) (*Hell and Wichmann, 1994*). In both *Otof* $^{C2C/+}$ and *Otof* $^{C2C/C2C}$ IHCs, the distance of a given otoferlin immunostaining spot to the closest spot of Vglut3 immunostaining had a different distribution than the distance of a given Vglut3 immunostaining spot from the closest Vglut3 immunostaining spot, with a distinctly smaller mode value (Kolmogorov-Smirnov test, p<10$^{-8}$ in the four examples shown in *Figure 3—figure supplement 2*). Moreover, the distributions of the distances between a given otoferlin spot and the closest Vglut3 spot were similar in *Otof* $^{C2C/+}$ and *Otof* $^{C2C/C2C}$ IHCs (Kolmogorov-Smirnov test, p>0.05 in the four examples shown in *Figure 3—figure supplement 2*). This distribution pooled from 7 *Otof* $^{C2C/C2C}$ IHCs sections peaked at 124 nm, a value smaller by a factor of 2.3 than the peak of the distribution of nearest neighbor distances between Vglut3 spots (285 nm), and probably corresponding to the resolution of our imaging conditions. These results suggest that otoferlin associates correctly with synaptic vesicles in *Otof* $^{C2C/C2C}$ IHCs.

We then studied the ultrastructure of IHC ribbon synapses by 3D electron tomography (*Figure 4A*). We reconstructed the ribbon synapses of IHCs located in the cochlear apical turn from seven *Otof* $^{C2C/+}$ and ten *Otof* $^{C2C/C2C}$ mice, on P17 (*Figure 4—video 1,2*). Synaptic vesicles were classified into three different pools, according to their position relative to the presynaptic plasma membrane and the ribbon: (i) ribbon-associated vesicles with centers lying within 40 nm of the presynaptic plasma membrane were classified as the presumptive readily releasable pool (RRP); (ii) vesicles lying within 80 nm of the ribbon but not apposed to the presynaptic plasma membrane were classified as the ribbon-attached pool (RAP), and (iii) vesicles located between 80 nm and 350 nm from the ribbon surface and not apposed to the presynaptic plasma membrane comprised the outlying pool (OP) (*Figure 4A*) (*Lenzi et al., 1999; Kantardzhieva et al., 2013*). In *Otof* $^{C2C/C2C}$ ribbons (*n* = 10), the RRP, RAP, and OP contained 14.0 ± 0.8, 42.4 ± 5.0, and 32.9 ± 3.6 synaptic vesicles per ribbon synapse, respectively (*Figure 4B*). These values are similar to those previously reported for wild-type mice and other species (*Lenzi et al., 1999; Schnee et al., 2005; Kantardzhieva et al., 2013; Vogl et al., 2015*), and are consistent with the pool sizes we measured in *Otof* $^{C2C/+}$ IHCs (*n* = 7; RRP, 15.0 ± 0.8; RAP, 41.0 ± 2.8; OP, 33.9 ± 2.0; p>0.4 for each pool). Estimated vesicle densities in the RAP and OP (see Materials and methods) were similar in *Otof* $^{C2C/+}$ and *Otof* $^{C2C/C2C}$ IHCs (*Figure 4C*; p>0.4 for each pool). The mean distance of RRP vesicles from the presynaptic plasma membrane was also similar in *Otof* $^{C2C/+}$ (27.8 ± 1.2 nm) and *Otof* $^{C2C/C2C}$ IHCs (24.3 ± 1.4 nm) (*Figure 4D*; p=0.1). However, the mean distance of RAP vesicles from the ribbon was slightly larger in *Otof* $^{C2C/C2C}$ IHCs (46.6 ± 1.5 nm) than in *Otof* $^{C2C/+}$ IHCs (39.5 ± 1.7 nm) (*Figure 4D*, p<0.01). The normal expression of otoferlin and the well-preserved structure of the ribbon synapse in *Otof* $^{C2C/C2C}$ IHCs rendered this mouse model suitable for assessment of the roles of otoferlin in vesicle fusion and in synaptic vesicle pool replenishment.

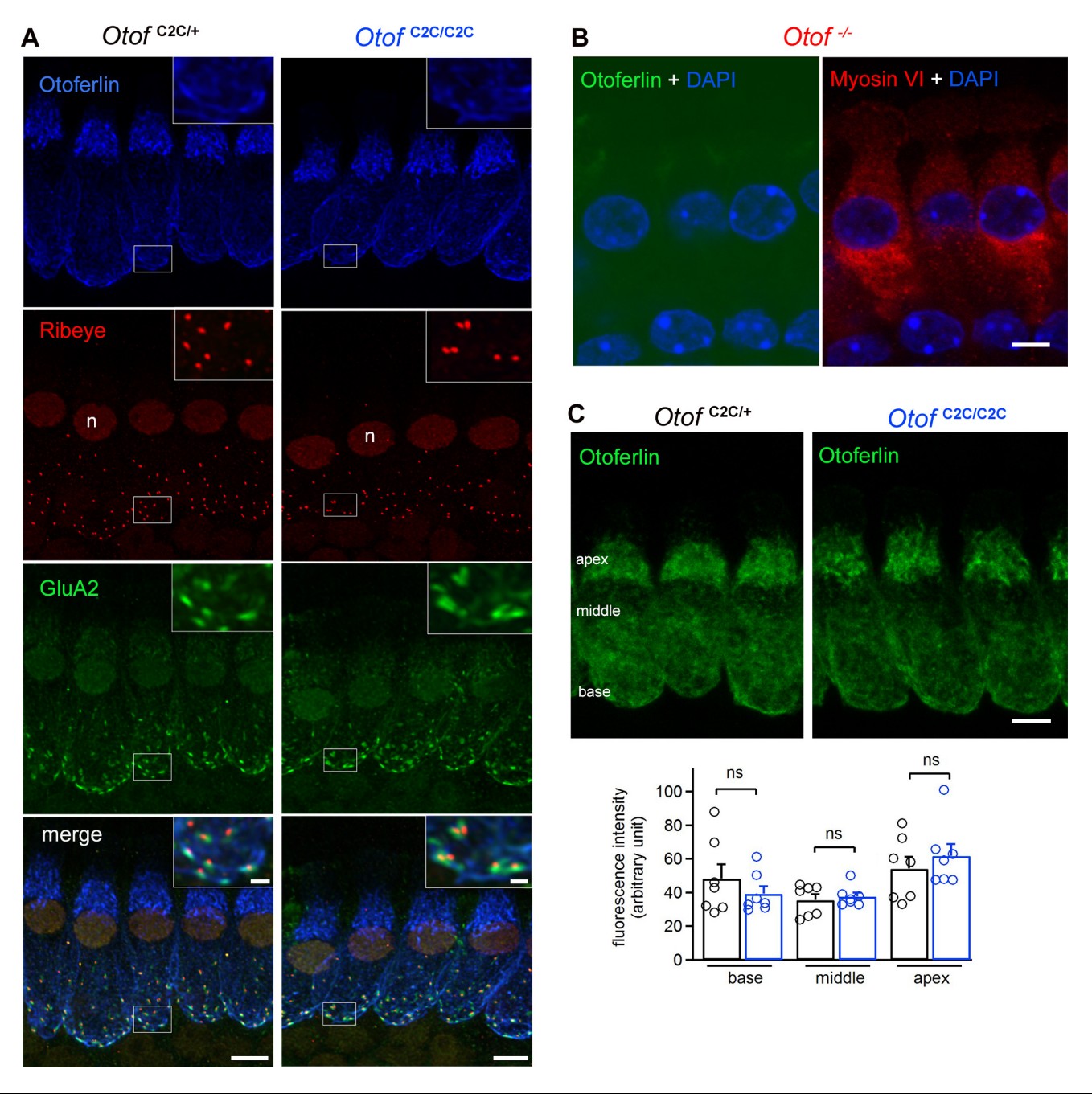

**Figure 3.** Normal expression and subcellular distribution of otoferlin in *Otof* C2C/C2C IHCs. (**A**) Confocal microscopy images of IHCs from whole-mount preparations of the organ of Corti from P15 *Otof* C2C/+ and *Otof* C2C/C2C mice triple-labeled for otoferlin (blue), ribeye (red), and postsynaptic GluA2 receptors (green). Insets: detailed views of the boxed synaptic areas. n: cell nucleus. Scale bar: 5 μm. Inset scale bar: 1 μm. (**B**) Confocal microscopy image of IHCs from a whole-mount preparation of an organ of Corti from a P15 *Otof* -/- mouse triple-labeled for otoferlin (green), the hair cell marker myosin VI (red), and the cell nucleus marker DAPI (blue). Note that the luminosity of the green channel (otoferlin) has been enhanced to show the absence of otoferlin expression in *Otof* -/- mice. Scale bar: 5 μm. (**C**) *Top*: Summed projected *z*-stack confocal microscopy images of IHCs from whole-mount preparations of organs of Corti from P15 *Otof* C2C/+ and *Otof* C2C/C2C mice labeled for otoferlin (green). Scale bar: 5 μm. *Bottom*: Quantification of otoferlin fluorescence in *Otof* C2C/+ (n = 51 cells in 7 mice) and *Otof* C2C/C2C IHCs (n = 51 cells from 7 mice) at the apex, middle, and the base of IHCs. Data information: in (**C**), data are presented as the mean ± SEM. ns, not significant (Student's *t*-test with Welch correction).
DOI: https://doi.org/10.7554/eLife.31013.005

The following figure supplements are available for figure 3:

**Figure supplement 1.** Normal number of ribbon synapses in *Otof* C2C/C2C IHCs.

*Figure 3 continued on next page*

*Figure 3 continued*

DOI: https://doi.org/10.7554/eLife.31013.006

**Figure supplement 2.** Normal colocalization of otoferlin and Vglut3 in *Otof*<sup>C2C/C2C</sup> IHCs.

DOI: https://doi.org/10.7554/eLife.31013.007

## The Ca$^{2+}$ sensitivity of RRP vesicle fusion is affected in *Otof*<sup>C2C/C2C</sup> IHCs

Deficient IHC synaptic exocytosis may be due to a failure of synaptic vesicle fusion, vesicle pool replenishment, or endocytosis and synaptic vesicle reformation. We sought to identify the steps of

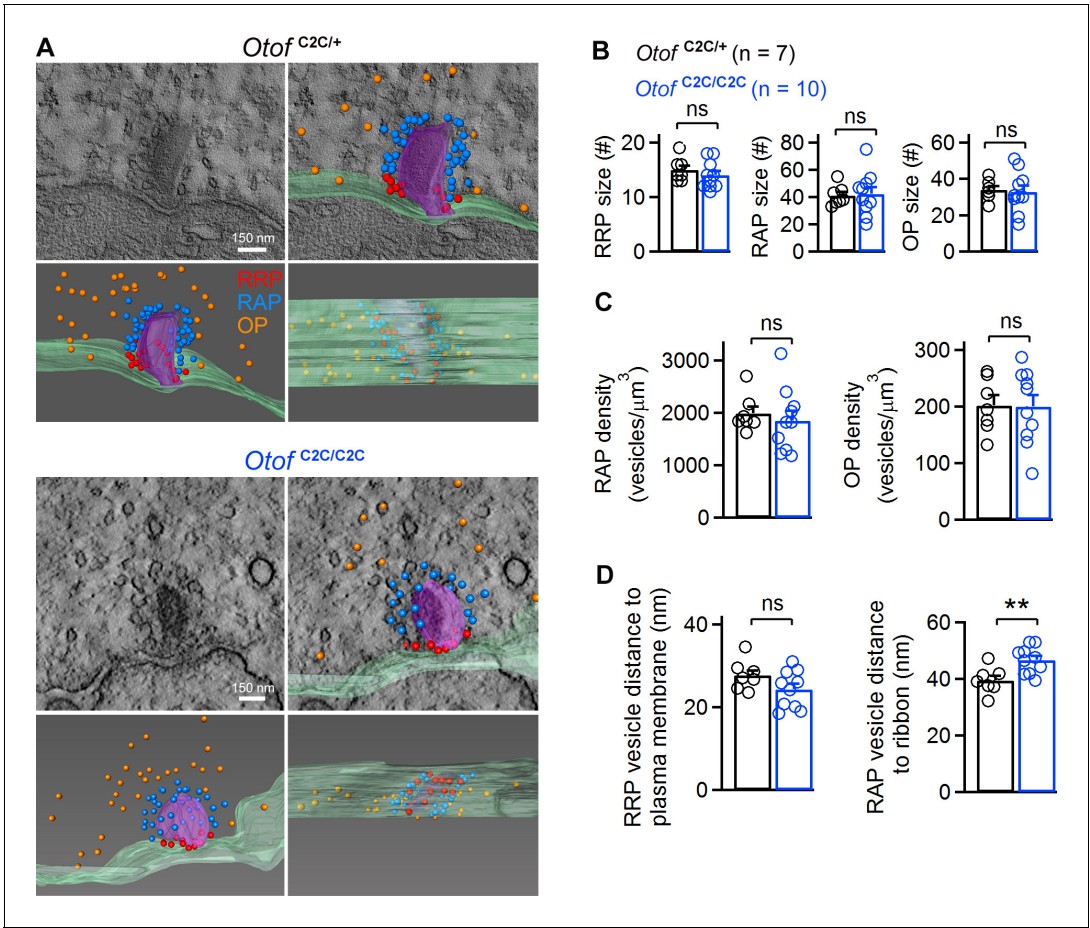

**Figure 4.** Ultrastructural analysis of the different vesicle pools in *Otof*<sup>C2C/C2C</sup> IHC ribbon synapses. (**A**) Representative transmission electron micrographs of ribbon synapses from an *Otof*<sup>C2C/+</sup> IHC (upper panels) and an *Otof*<sup>C2C/C2C</sup> IHC (lower panels). Side views of the electron tomographic reconstruction of *Otof*<sup>C2C/+</sup> and *Otof*<sup>C2C/C2C</sup> ribbon synapses are overlaid on the 3D tomograms. Renderings show the presynaptic plasma membrane (green) relative to the ribbon (purple), vesicles of the presumptive readily-releasable pool (RRP) in red, of the ribbon-associated pool (RAP) in blue, and of the outlying pool (OP) in orange. The ribbon was made semitransparent for visualization of its 3D shape and of the vesicles behind it. (**B**) Mean numbers of vesicles in the RRP, RAP, and OP of reconstructed *Otof*<sup>C2C/+</sup> (*n* = 7) and *Otof*<sup>C2C/C2C</sup> (*n* = 10) ribbon synapses. (**C**) Mean vesicle densities in the RAP and OP of reconstructed *Otof*<sup>C2C/+</sup> (*n* = 7) and *Otof*<sup>C2C/C2C</sup> (*n* = 10) ribbon synapses. (**D**) Mean Euclidean distance from RRP vesicle centers to the plasma membrane and from RAP vesicle centers to the ribbon in reconstructed *Otof*<sup>C2C/+</sup> (*n* = 7) and *Otof*<sup>C2C/C2C</sup> (*n* = 10) ribbon synapses. Data information: In (**B–D**), data are presented as the mean ± SEM. **$p < 0.01$, ns not significant (Student's *t*-test with Welch correction).

DOI: https://doi.org/10.7554/eLife.31013.008

The following videos are available for figure 4:

**Figure 4—video 1.** Example of a ribbon synapse from an *Otof*<sup>C2C/+</sup> mouse reconstructed by electron tomography

DOI: https://doi.org/10.7554/eLife.31013.009

**Figure 4—video 2.** Example of a ribbon synapse from an *Otof*<sup>C2C/C2C</sup> mouse reconstructed by electron tomography

DOI: https://doi.org/10.7554/eLife.31013.010

the IHC synaptic vesicle cycle affected in $Otof^{C2C/C2C}$ mice, by monitoring depolarization-evoked membrane capacitance changes ($\Delta C_m$) in the IHCs of P15-P18 mice, under various conditions of stimulation. We varied $Ca^{2+}$ influx through the L-type voltage-gated $Ca^{2+}$ channels by modifying IHC depolarization level, depolarization duration, or extracellular $Ca^{2+}$ concentration.

We first measured $Ca^{2+}$ currents ($I_{Ca}$) and the corresponding $\Delta C_m$ in response to depolarizations of various amplitudes (from a holding membrane potential of $-95$ mV to potentials between $-65$ mV and $+35$ mV), each lasting 20 ms, during which synaptic exocytosis mostly reflects the fusion of RRP vesicles (*Figure 5A1–A2* and *Figure 5—figure supplement 1A*) (*Moser and Beutner, 2000*). Plots of $I_{Ca}$ amplitude as a function of membrane potential were similar between $Otof^{+/+}$ ($n = 8$) and $Otof^{C2C/+}$ IHCs ($n = 11$; two-way-ANOVA, p=0.12), and between $Otof^{C2C/+}$ ($n = 11$) and $Otof^{C2C/C2C}$ IHCs ($n = 15$; two-way-ANOVA, p=0.4). The $I_{Ca}$ plots had the typical inverted bell shape, with a negative peak at around $-10$ mV (*Figure 5A2*, top and *Figure 5—figure supplement 1B*, top). The corresponding $\Delta C_m$ also peaked at about $-10$ mV (*Figure 5A2*, bottom and *Figure 5—figure supplement 1B*, bottom). In $Otof^{+/+}$ and $Otof^{C2C/+}$ IHCs, the evoked $\Delta C_m$ was identical (two-way-ANOVA, p=0.93; *Figure 5—figure supplement 1B*, bottom), and we used $Otof^{C2C/+}$ IHCs as control cells (see Materials and methods). In comparisons of release in $Otof^{C2C/C2C}$ and $Otof^{C2C/+}$ IHCs, the $\Delta C_m$ evoked by stepwise depolarization from a holding membrane potential of $-95$ mV to potentials between $-65$ mV and $-30$ mV, characterized by a low opening probability for $Ca^{2+}$-channels and a strong $Ca^{2+}$ influx-driving force, was similar. However, for depolarization to $-30$ mV and beyond (see dashed line in *Figure 5A2*), corresponding to a high open probability of $Ca^{2+}$-channels and a decreasing driving force for $Ca^{2+}$ entry, $\Delta C_m$ was significantly smaller in $Otof^{C2C/C2C}$ IHCs than in $Otof^{C2C/+}$ IHCs (two-way-ANOVA, $p<10^{-4}$). A 1.6-fold decrease was observed for depolarization to $-10$ mV. Assuming a membrane capacitance of 45 aF for a single synaptic vesicle (*Neef et al., 2007*), this corresponds to a difference of 8.3 vesicles/ms in vesicle fusion rate during the 20 ms of depolarization between $Otof^{C2C/+}$ IHCs (21.7 vesicles/ms) and $Otof^{C2C/C2C}$ IHCs (13.4 vesicles/ms). We then assessed the $Ca^{2+}$ efficiency of RRP release, by plotting $\Delta C_m$ as a function of $I_{Ca}$ amplitude for depolarization to membrane potentials of $-65$ mV to $-10$ mV, corresponding to the falling segment of the $Ca^{2+}$ current amplitude-voltage ($I_{Ca}/V_m$) curve (*Figure 5A3*), in which $Ca^{2+}$ currents display minimal contamination with residual unblocked $K^+$ currents. The $Ca^{2+}$ efficiency of RRP release, evaluated by plotting the mean derivative of $\Delta C_m$ against $I_{Ca}$, was lower (by a factor of 1.7) in $Otof^{C2C/C2C}$ IHCs (($3.9 \pm 0.7$) x $10^{-2}$ fF.pA$^{-1}$) than in $Otof^{C2C/+}$ IHCs (($6.6 \pm 0.4$) x $10^{-2}$ fF.pA$^{-1}$) (*Figure 5A3*; 95% confidence interval of the fitted slope).

We investigated the role of otoferlin in the kinetics of RRP synaptic vesicle fusion further, by analyzing the $\Delta C_m$ elicited by brief depolarizations, of 2 to 50 ms duration, to $-10$ mV (*Figure 5B1*), first in low intracellular $Ca^{2+}$-buffering conditions with an intracellular solution containing 0.5 mM EGTA. The presence of a vesicle fusion defect per se, independent of vesicle pool replenishment, could be inferred from the $\Delta C_m$ evoked by very short IHC depolarizations (<10 ms) (*Figure 5B2*). For depolarizations of 2 and 5 ms, the $\Delta C_m$ values in $Otof^{C2C/C2C}$ mice ($n = 16$) were about a third (p=0.01) and a half (p=0.02) those in $Otof^{C2C/+}$ mice ($n = 11$), respectively, indicating a direct effect of the $C_2C$ domain mutations on vesicle fusion. By fitting the average relationship between $\Delta C_m$ and depolarization duration ($\Delta t$) for the shortest depolarizations, we found that the slope of this fit, taken as an approximation of the $Ca^{2+}$ sensitivity of release, was lower (by a factor of 1.5) in $Otof^{C2C/C2C}$ IHCs (0.46 $\pm$ 0.04 fF.ms$^{-1}$) than in $Otof^{C2C/+}$ IHCs (0.69 $\pm$ 0.05 fF.ms$^{-1}$) (*Figure 5B2*; 95% confidence interval of the fitted slope). This result again suggests that the $C_2C$ domain mutations substantially decrease the $Ca^{2+}$-sensitivity of RRP vesicle fusion. We verified that the low vesicle fusion rate in $Otof^{C2C/C2C}$ IHCs was not due to a looser coupling of $Ca^{2+}$ channels with the release sites, by patch-clamping a subset of IHCs with an intracellular solution containing 5 mM EGTA, a high buffer concentration reported to limit intracellular $Ca^{2+}$ diffusion from its entry point, thereby reducing the fusion of vesicles lying more than a few tens of nm away from $Ca^{2+}$ channels (*Moser and Beutner, 2000*; *Spassova et al., 2004*; *Brandt et al., 2005*; *Levic et al., 2011*) (*Figure 5B3*). In $Otof^{C2C/+}$ IHCs, RRP release at intracellular EGTA concentrations of 0.5 mM ($n = 11$) and 5 mM ($n = 9$) was similar for depolarization durations between 2 and 10 ms, reflecting a tight coupling of $Ca^{2+}$ channels to RRP vesicles (*Figure 5B3*, top, two-way-ANOVA, p=0.3 for the 2–10 ms interval and p=$3\times10^{-4}$ for the 2–30 ms interval). In $Otof^{C2C/C2C}$ IHCs, RRP release at intracellular EGTA concentrations of 0.5 mM ($n = 16$) and 5 mM ($n = 10$) was also similar for depolarization durations between 2 and 10 ms (*Figure 5B3*, bottom; two-way-ANOVA, p=0.2 for the 2–30 ms interval), suggesting that the

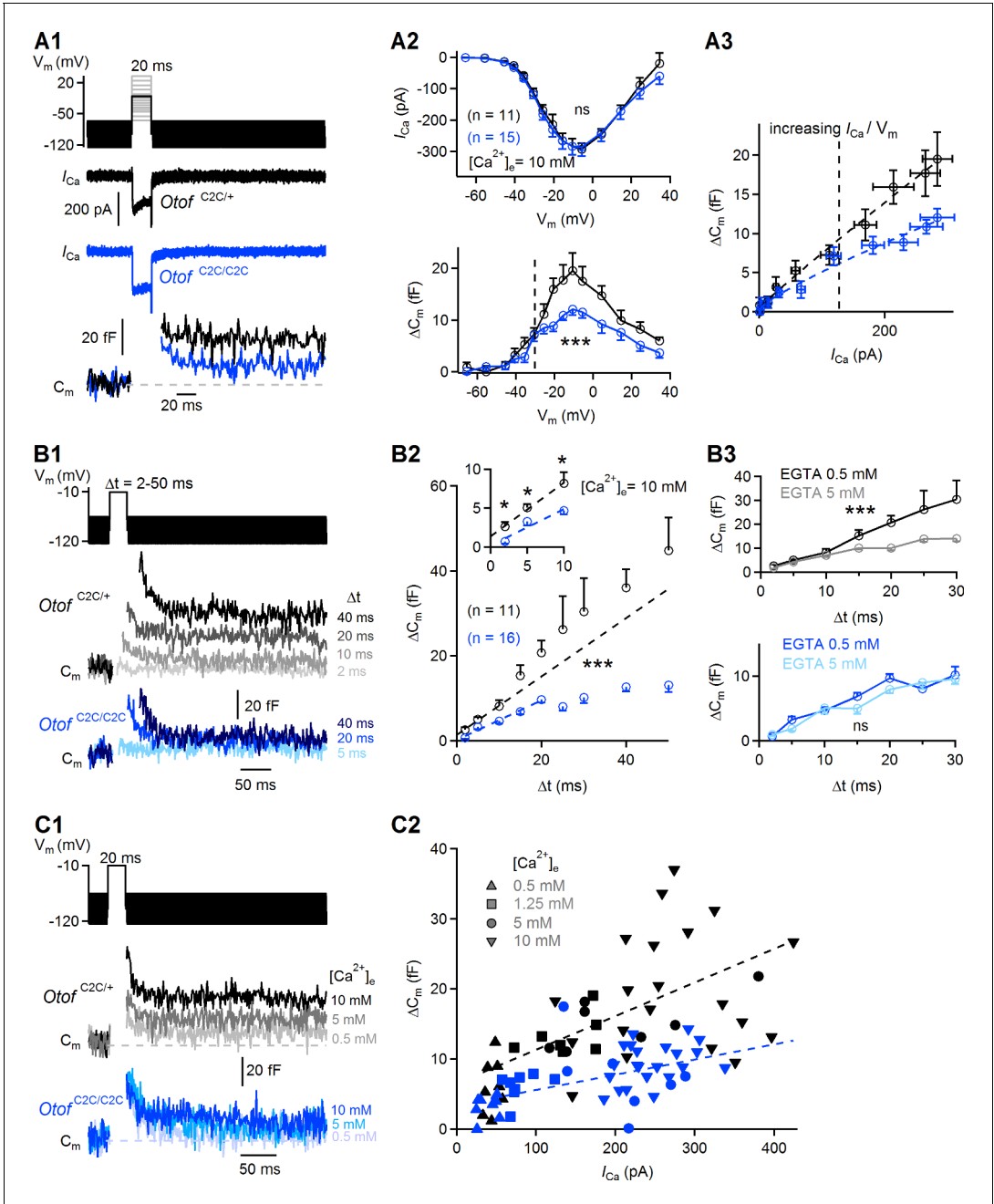

**Figure 5.** Ca$^{2+}$-dependence of the RRP vesicle fusion defect in *Otof*$^{C2C/C2C}$ IHCs. (**A1**) Protocol used to depolarize IHCs from −95 mV to potentials between −65 to +35 mV (top). Examples of Ca$^{2+}$ currents ($I_{Ca}$) (middle) and corresponding C$_m$ traces (bottom) for P15-P18 *Otof*$^{C2C/+}$ and *Otof*$^{C2C/C2C}$ IHCs after 20 ms of depolarization to −10 mV. (**A2**) Mean Ca$^{2+}$ current amplitudes ($I_{Ca}$) (top) and ΔC$_m$ (bottom) for P15-P18 *Otof*$^{C2C/+}$ and *Otof*$^{C2C/C2C}$ IHCs after 20 ms of depolarization to potentials between −65 mV to +35 mV. The vertical dashed line indicates the −30 mV voltage point. (**A3**) Mean ΔC$_m$ values plotted against the Ca$^{2+}$ currents elicited by depolarizing steps to potentials underlying the falling segment of the $I_{Ca}$/V$_m$ curve (−65 mV to −10 mV), corresponding to increasing Ca$^{2+}$ currents. The vertical dashed line indicates the −30 mV voltage point. The *Otof*$^{C2C/+}$ and *Otof*$^{C2C/C2C}$ ΔC$_m$ data were fitted with a power function, yielding an exponent of 0.94 and 0.83, respectively. (**B1**) Protocol used to depolarize IHCs from −95 mV to −10 mV for voltage steps of different durations from 2 ms to 50 ms (top). Corresponding example C$_m$ traces from P15-P18 *Otof*$^{C2C/+}$ and *Otof*$^{C2C/C2C}$ IHCs (bottom). The example traces for each genotype come from the same patch-clamped IHC. (**B2**) Kinetics of Ca$^{2+}$-dependent exocytosis in P15-P18 *Otof*$^{C2C/+}$ and *Otof*$^{C2C/C2C}$ IHCs for voltage steps of 2 ms to 50 ms. Mean ΔC$_m$ is plotted against the duration of the depolarization to −10 mV (Δt). The inset shows the detail for Δt values between 2 ms and 10 ms. For the 2 ms and 5 ms depolarizations, five repetitions of the recordings were averaged, to increase the signal-to-noise ratio. The decrease in Ca$^{2+}$-sensitivity of RRP vesicle fusion was evaluated by fitting the ΔC$_m$ versus Δt plots with a line for Δt between 2 and 10 ms in *Otof*$^{C2C/+}$ IHCs and for Δt between 2 and 20 ms in *Otof*$^{C2C/C2C}$ IHCs. The *Otof*$^{C2C/+}$ fit was plotted for durations greater than 10 ms, to illustrate the onset of the second component of release corresponding to the initiation of vesicle pool replenishment.

*Figure 5 continued on next page*

Figure 5 continued

(B3) We evaluated the coupling of voltage-gated Ca$^{2+}$ channels to RRP vesicles, by setting the intracellular EGTA concentration to 5 mM in *Otof*$^{C2C/+}$ IHCs (gray, *n* = 9) and in *Otof*$^{C2C/C2C}$ IHCs (light blue, *n* = 10). The data for an intracellular EGTA concentration of 0.5 mM are as in (B2). (C1) Protocol used to depolarize IHCs from −95 mV to −10 mV for 20 ms with different extracellular Ca$^{2+}$ concentrations (top). Example C$_m$ traces from P15-P18 *Otof*$^{C2C/+}$ and *Otof*$^{C2C/C2C}$ IHCs for different extracellular Ca$^{2+}$ concentrations (bottom). Each example C$_m$ trace for a given genotype was obtained from a different IHC. (C2) ΔC$_m$ values plotted against the Ca$^{2+}$ currents elicited at different extracellular Ca$^{2+}$ concentrations ([Ca$^{2+}$]$_e$) in *Otof*$^{C2C/+}$ and *Otof*$^{C2C/C2C}$ P15-P18 IHCs. Dashed lines show linear fits to the data. Data information: In (A2, B2–B3), data are presented as the mean ± SEM. ***p<0.001, ns not significant (two-way-ANOVA). In (B2, inset), *p<0.05 (Student's *t*-test with Welch correction). In A1, example Ca$^{2+}$ traces are corrected for linear leak conductance, leading to a subtraction of the sinusoidal signal. In (A1, B1, C1), the raw C$_m$ traces are shown.
DOI: https://doi.org/10.7554/eLife.31013.011

The following figure supplement is available for figure 5:

**Figure supplement 1.** Ca$^{2+}$-dependence of RRP vesicle fusion is normal in *Otof*$^{C2C/+}$ IHCs.
DOI: https://doi.org/10.7554/eLife.31013.012

spatial coupling between voltage-gated Ca$^{2+}$ channels and RRP vesicles is not modified by the otoferlin C$_2$C domain mutations. Remarkably, unlike the exocytotic response of *Otof*$^{C2C/+}$ IHCs, which did not plateau for depolarizations lasting up to 50 ms, that of *Otof*$^{C2C/C2C}$ IHCs rapidly saturated for depolarizations lasting 15–20 ms (*Figure 5B2*), and was insensitive to 5 mM intracellular EGTA (*Figure 5B3*), suggesting that vesicle pool replenishment at the release sites was also impaired in these cells.

We then characterized the defect in Ca$^{2+}$ sensitivity by varying the amplitude of Ca$^{2+}$ entry through the voltage-gated Ca$^{2+}$ channels using a set of different extracellular Ca$^{2+}$ concentrations ([Ca$^{2+}$]$_e$ = 0.5 mM, 1.25 mM, 5 mM, 10 mM) during depolarizations of the IHCs for 20 ms to −10 mV (*Figure 5C1*). Similar maximal voltage-gated Ca$^{2+}$ currents were obtained, ranging from 45 ± 3 pA ([Ca$^{2+}$]$_e$ = 0.5 mM) to 268 ± 18 pA ([Ca$^{2+}$]$_e$ = 10 mM) in *Otof*$^{C2C/+}$ IHCs (*n* = 43) and from 40 ± 4 pA ([Ca$^{2+}$]$_e$ = 0.5 mM) to 246 ± 9 pA ([Ca$^{2+}$]$_e$ = 10 mM) in *Otof*$^{C2C/C2C}$ IHCs (*n* = 47; p>0.2 for both comparisons). However, for any given level of Ca$^{2+}$ current, synaptic release in *Otof*$^{C2C/C2C}$ IHCs was only about half that in *Otof*$^{C2C/+}$ IHCs. Moreover, the Ca$^{2+}$ sensitivity of release, evaluated by fitting a linear function to the ΔC$_m$/I$_{Ca}$ curve, was lower (by a factor of 2.2) in *Otof*$^{C2C/C2C}$ IHCs ((2.2 ± 0.5) x 10$^{-2}$ fF.pA$^{-1}$) than in *Otof*$^{C2C/+}$ IHCs ((4.8 ± 1.0) x 10$^{-2}$ fF.pA$^{-1}$) (*Figure 5C2*; 95% confidence interval of the fitted slope). Together, these results support the notion that C$_2$C domain mutations decrease the Ca$^{2+}$-dependent efficiency and kinetics of RRP vesicle fusion.

## Vesicle release in response to intracellular Ca$^{2+}$ uncaging is delayed and slower in *Otof*$^{C2C/C2C}$ IHCs

We studied the Ca$^{2+}$ dependence of the C$_2$C-mutated otoferlin independently of the Ca$^{2+}$ channels, by monitoring ΔC$_m$ upon intracellular Ca$^{2+}$ uncaging, using a single high-energy UV flash delivered to P14-P16 IHCs loaded with Ca$^{2+}$-saturated DM-nitrophen (*Heidelberger et al., 1994*; *Beutner et al., 2001*; *Vincent et al., 2014*). This protocol, which triggered a rapid and global increase in intracellular Ca$^{2+}$ concentration, from a few tens of nM to up to 59 ± 7 μM (*n* = 7) (*Vincent et al., 2015*), led to a total synaptic release that was similar in *Otof*$^{C2C/+}$ IHCs (2.25 ± 0.18 pF; *n* = 18) and *Otof*$^{C2C/C2C}$ IHCs (2.19 ± 0.22 pF; *n* = 19; Mann-Whitney test, p=0.4; *Figure 6A*), corresponding to about 50 000 vesicles for both genotypes. This result suggests that all the vesicles in *Otof*$^{C2C/C2C}$ IHCs can fuse when exposed to high Ca$^{2+}$ concentrations throughout the entire vesicle cycle pathway. We then analyzed the kinetics of ΔC$_m$ in the first few milliseconds after the UV flash in *Otof*$^{C2C/C2C}$ IHCs. The most striking features were the delayed onset of synaptic release and the longer time required to reach the maximal rate of release in *Otof*$^{C2C/C2C}$ IHCs (7.3 ± 1.2 ms and 52.1 ± 9.2 ms, respectively) than in *Otof*$^{C2C/+}$ IHCs (4.2 ± 0.7 ms and 24.3 ± 2.7 ms, respectively; Mann-Whitney test, p=0.02 and p=0.001, respectively; *Figure 6B–C*). The maximal rate of release was also lower (by a factor of 1.6) in *Otof*$^{C2C/C2C}$ IHCs (67 ± 14 fF.ms$^{-1}$, corresponding to 1490 ± 310 vesicles/ms) than in *Otof*$^{C2C/+}$ IHCs (107 ± 23 fF.ms$^{-1}$, corresponding to 2380 ± 510 vesicles/ms; Mann-Whitney test, p=0.03; *Figure 6C*).

We estimated the Ca$^{2+}$ sensitivity of vesicle fusion in *Otof*$^{C2C/+}$ and *Otof*$^{C2C/C2C}$ IHCs, by simultaneously monitoring ΔC$_m$ and intracellular Ca$^{2+}$ variations in IHCs loaded with the low-affinity

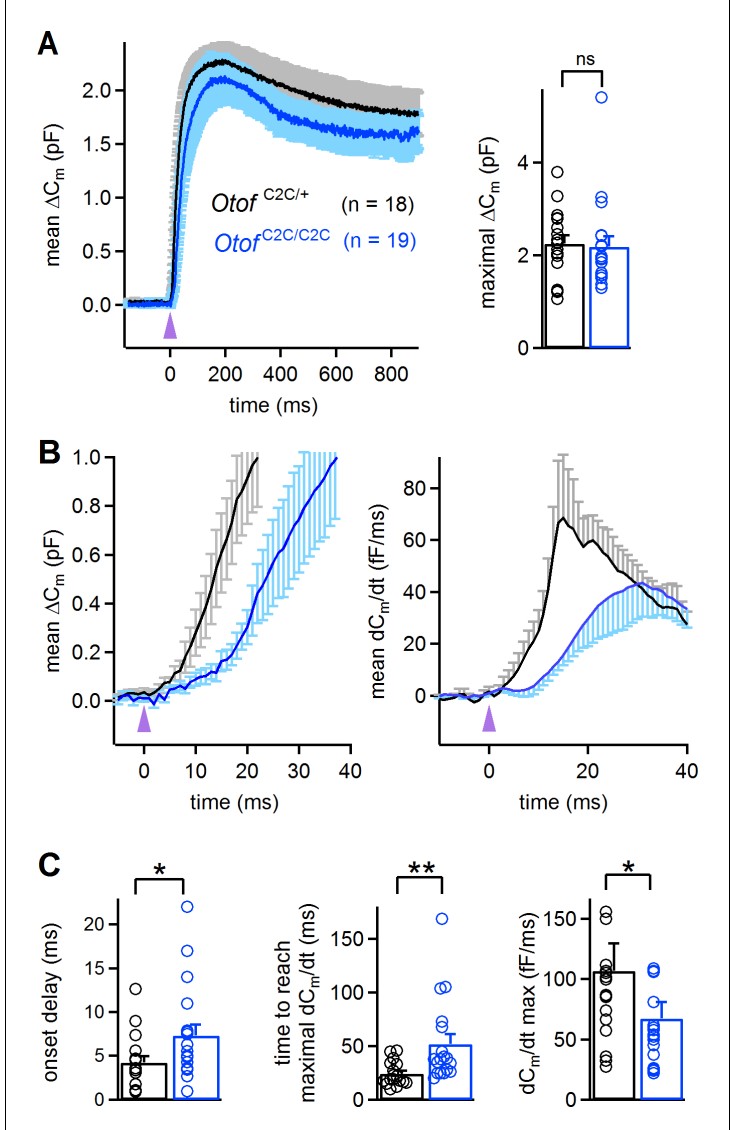

**Figure 6.** Abnormal kinetics of exocytosis evoked by intracellular Ca$^{2+}$ uncaging in *Otof* $^{C2C/C2C}$ IHCs. **(A)** *Left*: Mean ΔC$_m$ response curves in P14-P16 *Otof* $^{C2C/+}$ and *Otof* $^{C2C/C2C}$ IHCs after a single UV flash (purple arrowhead) uncaging intracellular Ca$^{2+}$. *Right*: Bar graph showing mean and individual maximal ΔC$_m$ values. **(B)** *Left*: Close-up of the wave shown in A, focusing on the first 40 ms after the UV flash (purple arrowhead). *Right*: Mean time derivative of the C$_m$ response curves in the first 40 ms following the UV flash. **(C)** Bar graphs of the mean and individual onset delay of the ΔC$_m$ (left), time taken to reach the maximal release rate (middle), and time derivative of release (right) in *Otof* $^{C2C/+}$ (*n* = 18) and *Otof* $^{C2C/C2C}$ (*n* = 19) IHCs. Data information: in **(A–C)**, data are presented as the mean ± SEM. *p<0.05, **p<0.01, ns not significant (Mann-Whitney test).

DOI: https://doi.org/10.7554/eLife.31013.013

fluorescent Ca$^{2+}$-indicator Oregon Green BAPTA 5N (OGB-5N) (*Figure 7A*). The use of the long-wavelength excitation dye (488 nm) OGB-5N allowed continuous fast monitoring of the Ca$^{2+}$ signal under confocal microscopy in IHCs (line scan at 1 kHz), no photolysis of DM-nitrophen being triggered by visible-light excitation at 488 nm (*Vincent et al., 2014*). In these recording conditions, the kinetics of the intracellular Ca$^{2+}$ increase in *Otof* $^{C2C/+}$ (t = 28.8 ± 1.7 ms; *n* = 6) and *Otof* $^{C2C/C2C}$ IHCs (26.6 ± 9.2 ms; *n* = 7) upon UV flash were similar (p=0.8), whereas the peak exocytosis rate occurred later in *Otof* $^{C2C/C2C}$ IHCs than in *Otof* $^{C2C/+}$ IHCs in agreement with the first set of Ca$^{2+}$ uncaging experiments (*Figures 6A* and *7A*). We estimated the cooperativity and Ca$^{2+}$ sensitivity of synaptic release in *Otof* $^{C2C/+}$ and *Otof* $^{C2C/C2C}$ IHCs, by fitting the relationship between the rate of

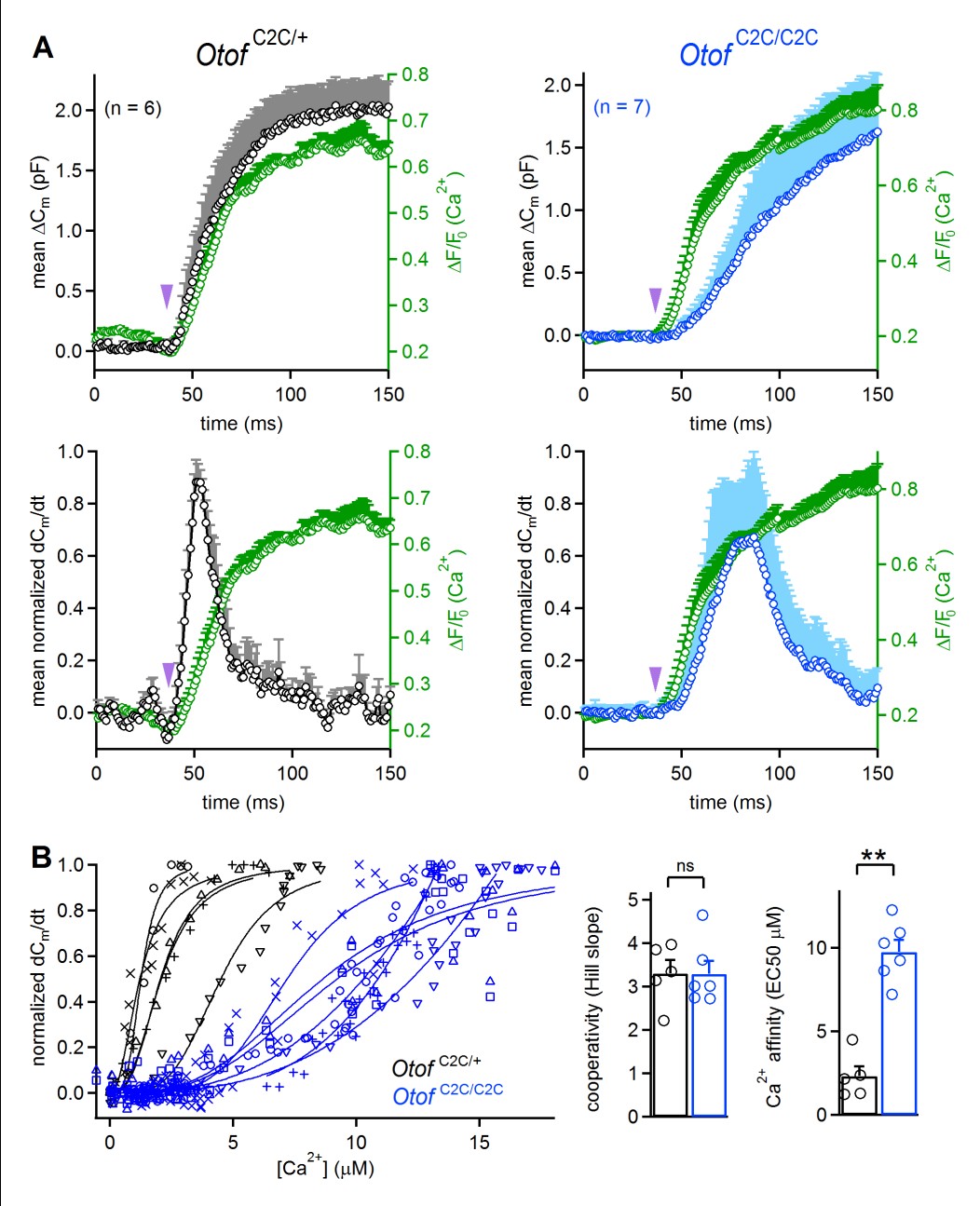

**Figure 7.** Reduced Ca$^{2+}$-sensitivity of exocytosis in *Otof* $^{C2C/C2C}$ IHCs. (A) Mean $\Delta C_m$ response curves (top) and normalized exocytosis rates ($dC_m/dt$) (bottom), with mean intracellular fluorescent Ca$^2$ signals of OGB-5N (green traces) recorded simultaneously after Ca$^{2+}$ uncaging (purple arrowhead) in P14-P16 *Otof* $^{C2C/+}$ (in black) and *Otof* $^{C2C/C2C}$ (in blue) IHCs. (B) *Left*: For each recorded *Otof* $^{C2C/+}$ (in black) and *Otof* $^{C2C/C2C}$ (in blue) IHC, the rate of exocytosis was plotted as a function of [Ca$^{2+}$]$_i$, and each data set was fitted with a Hill function $y = V_{max} \cdot x^n /(EC_{50}^n + x^n)$. *Right*: Mean cooperativity and Ca$^{2+}$-affinity of release in *Otof* $^{C2C/+}$ and *Otof* $^{C2C/C2C}$ IHCs. Note that for 3 out of 6 *Otof* $^{C2C/C2C}$ IHCs, the fit of the sigmoidal Hill function to the data was only partial. For those IHCs, the Ca$^{2+}$ affinity was estimated as the value of [Ca$^{2+}$] for which $dC_m/dt = 0.5$, rather than the EC$_{50}$ of the fitted Hill function, which was larger. Data information: in (B), one IHC of each genotype was removed because of the lack of experimental points for a reliable fitting with a Hill function. In (A–B), data are presented as the mean ± SEM. **p<0.01, ns not significant (Mann-Whitney test).

DOI: https://doi.org/10.7554/eLife.31013.014

exocytosis and the quantitatively calibrated fluorescent $Ca^{2+}$ signal for each cell with a Hill equation (*Figure 7B*, left). No major effect on the cooperativity of release was detected in *Otof*<sup>C2C/C2C</sup> IHCs (Hill coefficient value: $3.3 \pm 0.3$ for both genotypes; $p>0.9$; *Figure 7B*), whereas the $Ca^{2+}$ sensitivity of vesicle fusion was about four times lower in *Otof*<sup>C2C/C2C</sup> IHCs (EC50: $9.8 \pm 0.7$ μM) than that in *Otof*<sup>C2C/+</sup> IHCs (EC50: $2.3 \pm 0.6$ μM; $p=0.004$; *Figure 7B*). Taking into account the normal number of docked vesicles in *Otof*<sup>C2C/C2C</sup> IHCs (*Figure 4*) and the lower efficiency of $Ca^{2+}$-dependent exocytosis following the activation of their $Ca^{2+}$ channels (*Figure 5 C1–C2*), these $Ca^{2+}$ uncaging results are consistent with a deleterious effect of $C_2C$ domain mutations on the $Ca^{2+}$ affinity of otoferlin for triggering and setting the rate of RRP vesicle fusion.

## The $Ca^{2+}$ dependence of synaptic vesicle pool replenishment is affected in *Otof*<sup>C2C/C2C</sup> IHCs

It has been suggested that vesicle pool replenishment is otoferlin-dependent at the IHC ribbon synapse (*Pangrsic et al., 2010*), and $Ca^{2+}$-dependent thereafter (*Schnee et al., 2011b*; *Levic et al., 2011*). This prompted us to investigate the possible role of otoferlin as a $Ca^{2+}$ sensor for synaptic vesicle pool replenishment. We first determined the extent to which RRP replenishment was affected in *Otof*<sup>C2C/C2C</sup> IHCs, by performing paired-pulse experiments, using 20 ms-long pulses with various intervals between stimuli (*Figure 8A–B*). The paired-pulse ratio of release, used as a proxy for RRP replenishment, was much lower (by a factor of 2.2 at its minimum value) in *Otof*<sup>C2C/C2C</sup> IHCs ($n = 13$) than in *Otof*<sup>C2C/+</sup> IHCs ($n = 9$), for interpulse intervals of up to 500 ms (two-way ANOVA, $p<10^{-4}$). This result led us to conclude that the impairment of sustained release in *Otof*<sup>C2C/C2C</sup> IHCs is not merely a consequence of impaired RRP release; the recruitment of synaptic vesicles per se is also defective. However, the paired-pulse ratio recovered fully at interpulse intervals of about 1 s. We probed the response of *Otof*<sup>C2C/C2C</sup> IHCs ($n = 8$) to long depolarizations to −10 mV of up to 3 s and observed a much lower (by a factor of 5.3) rate of exocytosis than that measured in *Otof*<sup>C2C/+</sup> IHCs ($n = 8$; 80 fF/s versus 420 fF/s; two-way ANOVA, $p<10^{-4}$; *Figure 8C*).

We characterized the kinetics of synaptic vesicle pool replenishment further, by subjecting *Otof*<sup>C2C/C2C</sup> IHCs to periodic stimulation with 50 short (5 ms long) depolarizations to −10 mV, separated by 10 ms intervals (*Figure 8D–E*). This periodic stimulation induced a mean $\Delta C_m$ of 51 fF in *Otof*<sup>C2C/C2C</sup> IHCs (*Figure 8F*). This value exceeding that for RRP depletion (~20 fF), indicates the occurrence of vesicle pool replenishment. Assuming full depletion of the RRP by the train of depolarizations, we can estimate the rate of vesicle pool replenishment at about 1.5 vesicles/ms/IHC in *Otof*<sup>C2C/C2C</sup> IHCs, a much lower value (by a factor of 3.7) than obtained for *Otof*<sup>C2C/+</sup> IHCs (~5.6 vesicles/ms/IHC). The time course of synaptic exocytosis in *Otof*<sup>C2C/+</sup> IHCs during the train of successive depolarizations (*Figure 8D–F*) had three kinetic components, as previously described (*Schnee et al., 2011b*): a rapid depletion of the RRP during the first two to three stimulations, followed by a linear release component between the 3rd and 23rd (approximately) stimulation, with a superlinear release component beyond the 23rd stimulation. The linear release component has been suggested to reflect $Ca^{2+}$-dependent replenishment of the fusion site by vesicles from the recycling pool (*Rizzoli and Betz, 2005*), possibly corresponding to RAP vesicles, and the superlinear component observed at high $Ca^{2+}$ loads to reflect additional vesicle replenishment from the reserve pool, possibly involving OP vesicles transported to release sites by a mechanism dependent on $Ca^{2+}$-induced $Ca^{2+}$ release (*Schnee et al., 2011b*; *Castellano-Muñoz et al., 2016*). The linear replenishment component was less efficient (by a factor of 2.8) in *Otof*<sup>C2C/C2C</sup> IHCs ($n = 17$) than in *Otof*<sup>C2C/+</sup> IHCs ($n = 12$), at 0.6 fF and 1.7 fF per depolarization step, respectively. The onset of the superlinear release component has been shown to be $Ca^{2+}$-dependent (*Schnee et al., 2011b*). In *Otof*<sup>C2C/C2C</sup> IHCs, superlinear release was observed in only one of seventeen *Otof*<sup>C2C/C2C</sup> IHCs subjected to such trains of depolarization, whereas it was observed in six of the twelve *Otof*<sup>C2C/+</sup> IHCs studied and six of the eleven *Otof*<sup>+/+</sup> IHCs studied (Fisher's exact test, $p=0.01$). This almost total absence of superlinear release in *Otof*<sup>C2C/C2C</sup> IHCs was probably unlinked to intracellular $Ca^{2+}$ diffusion defects, because the patterns of calcium entry and diffusion imaged by high-speed swept-field confocal microscopy in IHCs loaded with OGB-5N were similar in *Otof*<sup>C2C/+</sup> ($n = 10–12$) and *Otof*<sup>C2C/C2C</sup> IHCs ($n = 6–10$). Thus, the $C_2C$ domain mutations did not affect the temporal and spatial dynamics of $Ca^{2+}$ entry and diffusion around the synaptic ribbon (*Figure 8—figure supplement*

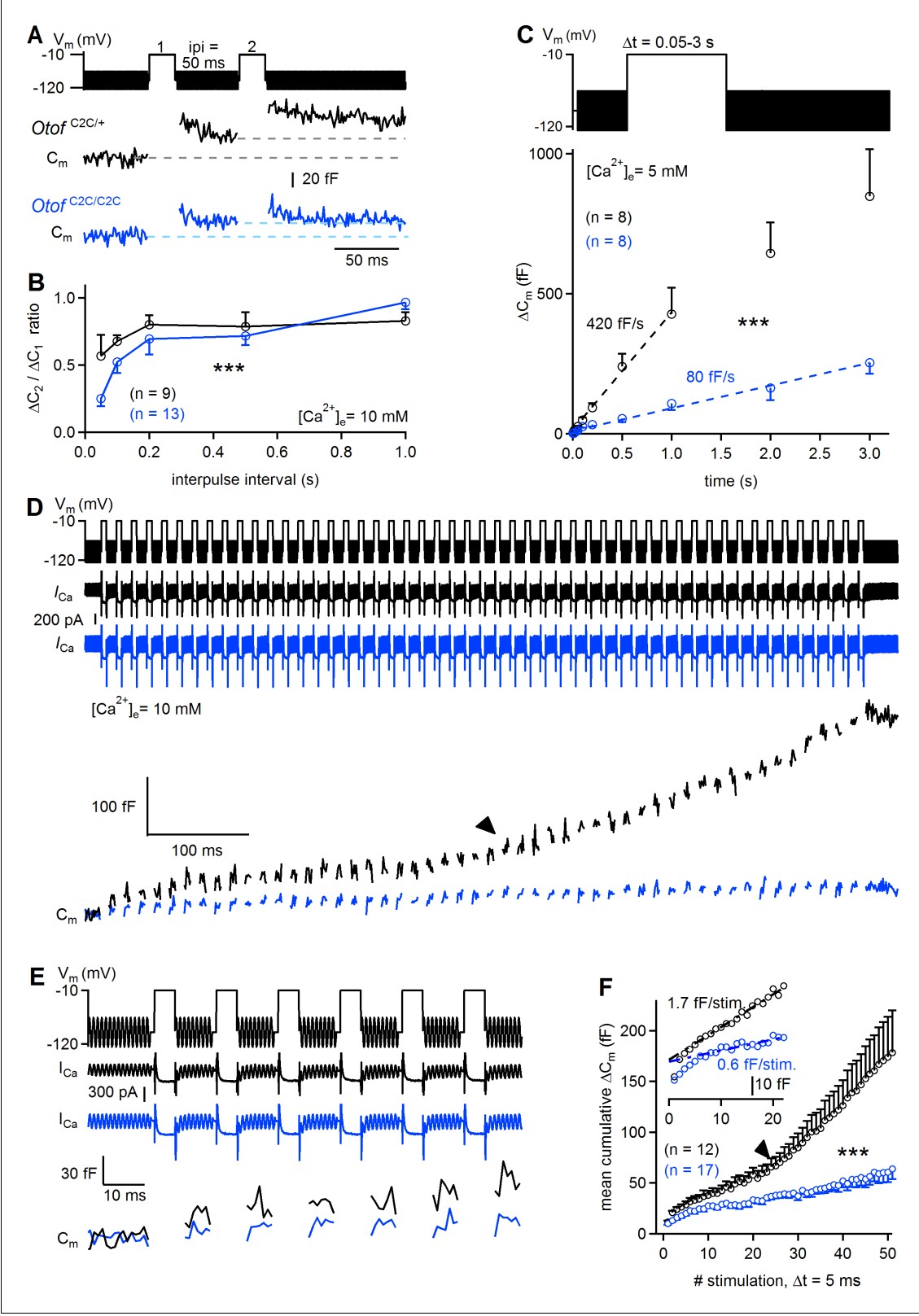

**Figure 8.** Reduced sustained release in *Otof* [C2C/C2C] IHCs. (**A**) Paired-pulse stimulation protocol, showing two consecutive 20 ms depolarizations from a holding potential of −95 mV to −10 mV (top), and example $C_m$ traces from *Otof* [C2C/+] and *Otof* [C2C/C2C] IHCs (bottom). (**B**) Mean $\Delta C_m$ ratio for the second depolarization relative to the first depolarization ($\Delta C_2/\Delta C_1$) as a function of interpulse interval (ipi = 50, 100, 200, 500, and 1000 ms), for P15-P18 *Otof* [C2C/+] and *Otof* [C2C/C2C] IHCs. (**C**) Kinetics of $Ca^{2+}$-dependent exocytosis for single depolarizations to −10 mV lasting between 50 ms and 3 s in

*Figure 8 continued on next page*

*Figure 8 continued*

P15-P18 IHCs. Mean $\Delta C_m$ values are plotted as a function of depolarization duration ($\Delta t$), together with linear fits for *Otof* [C2C/+] IHCs and *Otof* [C2C/C2C] IHCs. These recordings were carried out with an extracellular solution containing 5 mM $Ca^{2+}$. (D) Protocol used to elicit a train of 50 successive short depolarizations (duration 5 ms, interpulse interval 10 ms) to −10 mV (top). Example $I_{Ca}$ (middle) and corresponding $C_m$ traces (bottom) in *Otof* [C2C/+] and *Otof* [C2C/C2C] IHCs. Note the transition between the linear and superlinear components of release for the *Otof* [C2C/+] IHC (arrowhead). (E) Expanded view of the first 100 ms in D. (F) Plots of mean cumulative $\Delta C_m$ as a function of stimulus number in response to the train of 50 successive short depolarizations in *Otof* [C2C/+] and *Otof* [C2C/C2C] P15-P18 IHCs. For each depolarization, $\Delta C_m$ was evaluated in this particular case by averaging only the last 3 ms of the $C_m$ values of each interstimulus interval to prevent contamination by the initial peaks. Note the transition from the linear component to a superlinear component of release for *Otof* [C2C/+] IHCs (arrowhead). The inset is a magnification of the first 20 depolarizations, with linear fits to $\Delta C_m$ during successive depolarizations for *Otof* [C2C/+] and *Otof* [C2C/C2C] IHCs. Data information: in (B–C, F), data are presented as the mean ± SEM. ***p<0.001 (two-way-ANOVA). In (A, D, E) raw $C_m$ traces are shown. In (D, E) the $C_m$ transient change following each depolarization has been blanked for the sake of clarity. In (D), the example $I_{Ca}$ traces were not corrected for the linear leak conductance, leaving the sinusoidal $I_{Ca}$ component apparent.
DOI: https://doi.org/10.7554/eLife.31013.015

The following figure supplement is available for figure 8:

**Figure supplement 1.** Normal $Ca^{2+}$ entry, assessed by high-speed two-dimensional $Ca^{2+}$ imaging, in *Otof* [C2C/C2C] IHCs.
DOI: https://doi.org/10.7554/eLife.31013.016

*1*). These results suggest that otoferlin may also function as the $Ca^{2+}$ sensor triggering superlinear release.

## Endocytosis is normal in *Otof* [C2C/C2C] IHCs

It has also been suggested that otoferlin is involved in endocytosis (*Duncker et al., 2013*) and synaptic vesicle reformation (*Strenzke et al., 2016*). We therefore investigated endocytosis in *Otof*[C2C/C2C] IHCs, because a possible defect of endocytosis might account for the low rates of vesicle pool replenishment. In wild-type ribbon synapses, two modes of membrane retrieval following IHC depolarization, with different kinetics, have been described: a slow one involving linear $C_m$ decline upon short depolarizations (typically <50 ms in our recording conditions) and a fast one with an exponential decline upon longer depolarizations (typically >100 ms) (*Moser and Beutner, 2000*; *Beutner et al., 2001*; *Neef et al., 2014*). In *Otof* [C2C/+] and *Otof* [C2C/C2C] IHCs, both these modes of endocytosis were observed (*Figure 9A–B* and *Figure 9—figure supplement 1*). However, a 100 ms depolarization was sufficient to observe the exponential component of endocytosis in seven of the eight *Otof* [C2C/+] IHCs studied, but the same protocol elicited the exponential component in only one of the 10 *Otof* [C2C/C2C] IHCs recorded (Fisher's exact test, p=0.003). Increasing the duration of depolarization to 200–500 ms increased this proportion to six of the ten recorded *Otof* [C2C/C2C] IHCs. We therefore compared the slow and rapid endocytotic components between *Otof* [C2C/+] and *Otof* [C2C/C2C] IHCs with similar preceding exocytosis levels. In IHCs for which values of 20 to 150 fF had previously been recorded for exocytosis, the rate of the slow component was similar in *Otof* [C2C/+] (6.4 ± 0.8 fF.s$^{-1}$, $n$ = 8) and *Otof* [C2C/C2C] IHCs (5.3 ± 0.5 fF.s$^{-1}$, $n$ = 9; p=0.24; *Figure 9A*) and the $C_m$ traces systematically returned to the baseline $C_m$ value within 30 s after IHC depolarization. Likewise, when considering IHCs with an amplitude of the exponential component of endocytosis exceeding 100 fF, the time constant and amplitude of the exponential component of endocytosis were similar in *Otof* [C2C/+] (3.2 ± 0.5 s and 266 ± 44 fF, respectively, $n$ = 8) and *Otof* [C2C/C2C] IHCs (3.4 ± 0.3 s and 244 ± 34 fF, respectively, $n$ = 6; p>0.7 for both comparisons; *Figure 9B*), but the $C_m$ traces rarely reached the baseline $C_m$ value within 30 s after IHC depolarization. Finally, we further analyzed the $\Delta C_m$ measurements following a UV-flash. Upon $Ca^{2+}$ uncaging, the exocytotic peak was followed by a fast decline in $C_m$ that was probably due to endocytotic membrane retrieval, as previously described (*Beutner et al., 2001*). In these experimental conditions, endocytosis kinetics were also similar in *Otof* [C2C/+] ($n$ = 13) and *Otof* [C2C/C2C] ($n$ = 15; p=0.17; *Figure 9C*) IHCs. Together, our results indicate that endocytosis is unaffected in *Otof* [C2C/C2C] IHCs.

## The EPSC rate in IHC postsynaptic boutons is reduced in *Otof* [C2C/C2C] mice

The mean amplitude of excitatory postsynaptic currents (EPSCs) recorded at IHC postsynaptic boutons is known to be independent of presynaptic membrane voltage and $Ca^{2+}$ influx during sustained stimulation (*Goutman and Glowatzki, 2007*). We therefore expected the mean EPSC amplitude to

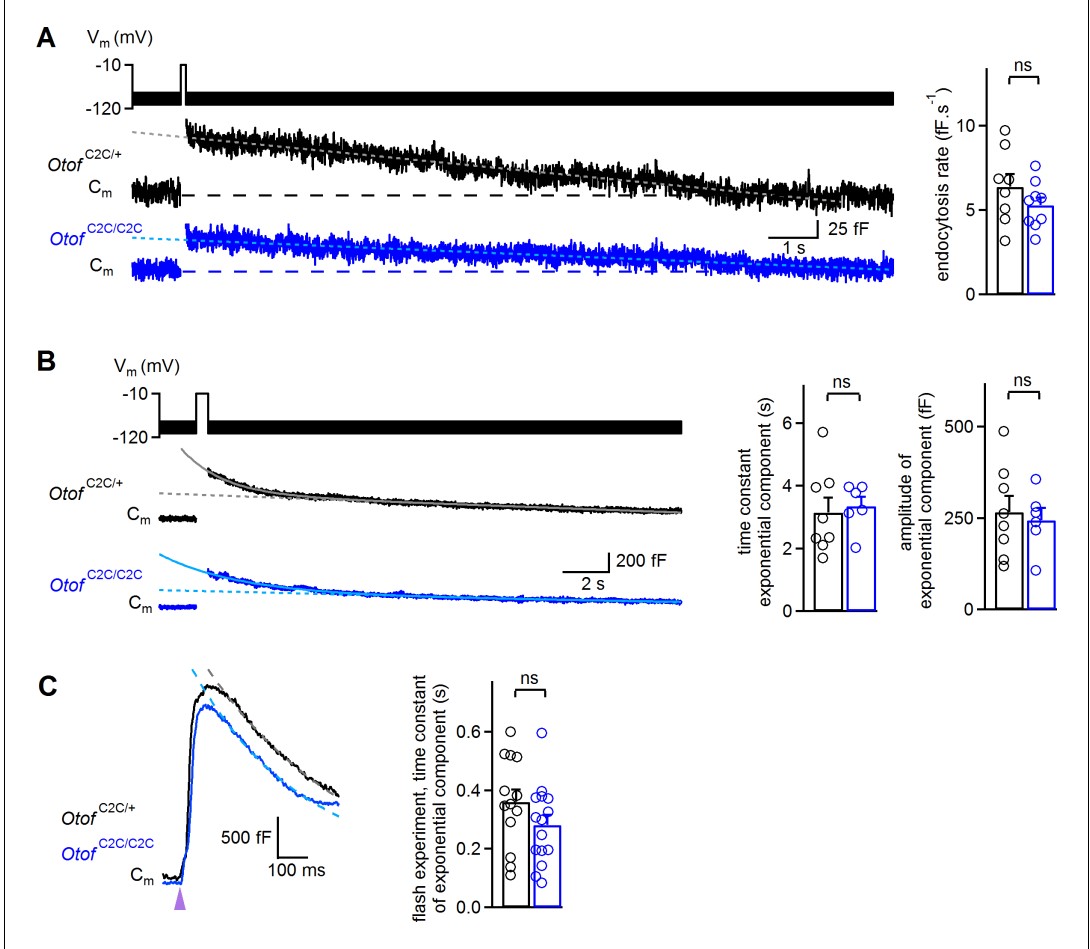

**Figure 9.** Normal endocytosis, assessed by prolonged $C_m$ measurements, in $Otof^{C2C/C2C}$ IHCs. (**A**) *Left*: Examples of $C_m$ traces, recorded over a period of 15 s, in response to a 100 ms depolarization to −10 mV from a holding potential of −95 mV, in P14-P17 $Otof^{C2C/+}$ and $Otof^{C2C/C2C}$ IHCs. Horizontal dashed lines indicate the baseline $C_m$. The decay phase of $C_m$ traces was fitted with a linear function (gray and light blue dotted lines). *Right*: Mean endocytosis rate in response to depolarizations lasting 20–100 ms in P14-P17 $Otof^{C2C/+}$ and $Otof^{C2C/C2C}$ IHCs. (**B**) *Left*: Examples of $C_m$ traces, recorded over a period of 20 s, in response to a 500 ms long depolarization to −10 mV from a holding potential of −95 mV, in $Otof^{C2C/+}$ and $Otof^{C2C/C2C}$ P14-P17 IHCs. The decay phase of $C_m$ traces was fitted with a monoexponential function added to a linear function (gray and light blue lines). Dotted lines correspond to the linear component of these fits. *Right*: Mean time constant and amplitude of the exponential component of $C_m$ decay in $Otof^{C2C/+}$ and $Otof^{C2C/C2C}$ IHCs. (**C**) *Left*: Example traces of the decay following exocytosis evoked by intracellular $Ca^{2+}$ uncaging (purple arrowhead) in $Otof^{C2C/+}$ and $Otof^{C2C/C2C}$ IHCs. The decay period was fitted with a monoexponential function (gray and light blue dashed lines). *Right*: Mean time constant of the exponential component of $C_m$ decay following exocytosis evoked by intracellular $Ca^{2+}$ uncaging. The data were collected from the IHCs used in *Figure 6*. Data information: in (**A–C**), data are presented as the mean ± SEM. ns not significant (Student's *t*-test with Welch correction). In (**A–B**), $C_m$ traces were 100 Hz low-pass filtered.

DOI: https://doi.org/10.7554/eLife.31013.017

The following figure supplement is available for figure 9:

**Figure supplement 1.** Example of a protocol used to probe endocytosis in an $Otof^{C2C/+}$ IHC and corresponding recorded traces.
DOI: https://doi.org/10.7554/eLife.31013.018

be unaffected in $Otof^{C2C/C2C}$ mice. We recorded EPSCs from single boutons on P8-P11, in the whole-cell patch-clamp configuration, as previously described (*Glowatzki and Fuchs, 2002*), eliciting IHC depolarization by increasing extracellular $K^+$ concentration from 5.8 mM to 25 mM (*Figure 10A*). EPSC frequency increased in both $Otof^{C2C/C2C}$ and $Otof^{+/+}$ IHCs. Mean EPSC amplitude and EPSC decay time constant did not differ significantly between $Otof^{C2C/C2C}$ (n = 8) and $Otof^{+/+}$ IHCs (n = 8; p>0.3; *Figure 10B*). The mean EPSC rate in the first 10 s following 25 mM KCl application was, however, slower in $Otof^{C2C/C2C}$ than in $Otof^{+/+}$ IHCs (by a factor of 3.3; *Figure 10C*, p=0.003), consistent with the smaller amplitude of the ABR wave-I and the presynaptic

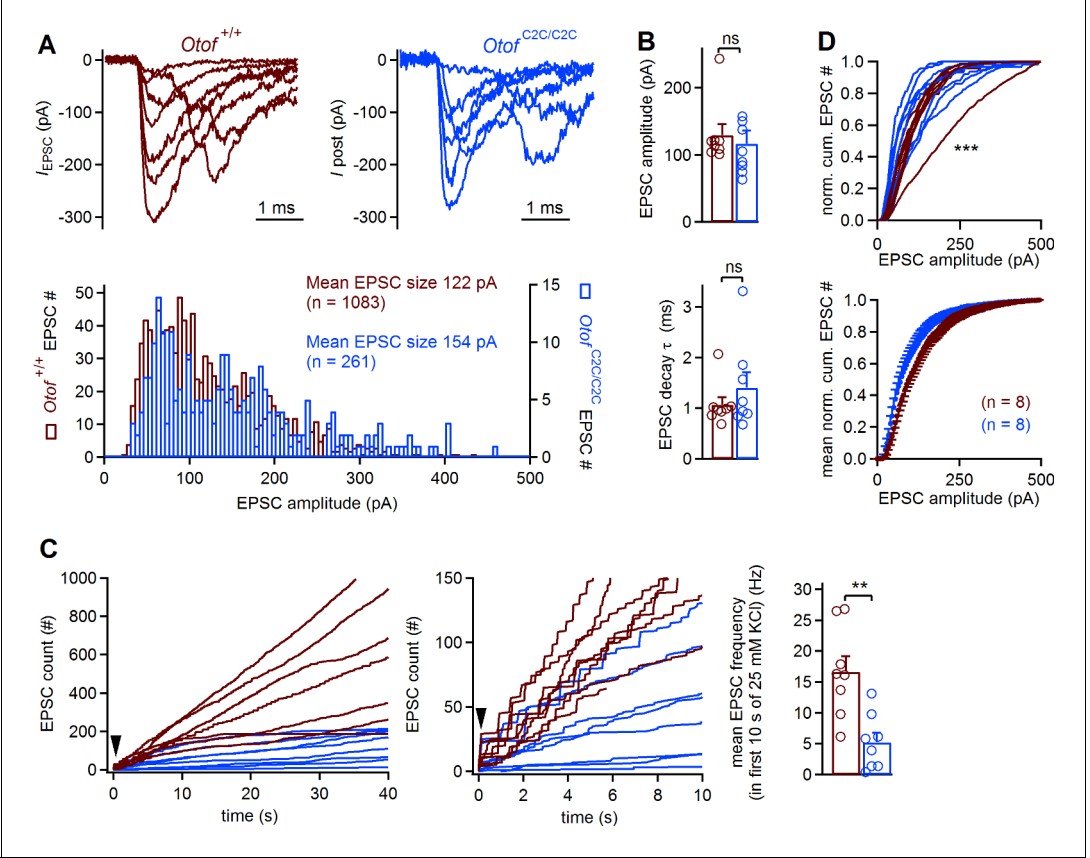

**Figure 10.** Lower EPSC rates in *Otof* <sup>C2C/C2C</sup> IHCs. (A) Superimposed example EPSC traces (*I*<sub>EPSC</sub>) recorded at *Otof* <sup>+/+</sup> and *Otof* <sup>C2C/C2C</sup> IHC afferent boutons on P8-P11, after extracellular application of 25 mM KCl (top), and histograms of EPSC amplitude distribution for the corresponding cells (bottom). (B) Bar graphs displaying the mean and individual EPSC amplitude and EPSC decay time constant (τ)in *Otof* <sup>+/+</sup> (*n* = 8) and *Otof* <sup>C2C/C2C</sup> (*n* = 8) IHCs. (C) *Left*: EPSC count as a function of time after the extracellular application of 25 mM KCl (indicated by an arrowhead) for all recorded *Otof* <sup>+/+</sup> and *Otof* <sup>C2C/C2C</sup> IHCs. *Middle*: Zoom on the first 10 s of recording. *Right*: Bar graph displaying mean EPSC rate in the first 10 s of the K<sup>+</sup> challenge, in *Otof* <sup>+/+</sup> and *Otof* <sup>C2C/C2C</sup> IHCs. (D) Mean and individual normalized cumulative EPSC number plotted against EPSC amplitude for *Otof* <sup>+/+</sup> and *Otof* <sup>C2C/C2C</sup> IHCs. Data information: in (B–D), data are presented as the mean ± SEM. In (B–C), **p<0.01, ns not significant (Student's *t*-test with Welch correction). In (D), ***p<0.001 (Kolmogorov-Smirnov test).

DOI: https://doi.org/10.7554/eLife.31013.019

exocytosis deficit in *Otof* <sup>C2C/C2C</sup> IHCs. Single-vesicle and multivesicular release events were observed in both *Otof* <sup>C2C/C2C</sup> and *Otof* <sup>+/+</sup> IHCs (*Figure 10A*). However, the distribution of EPSC amplitudes was more variable in *Otof* <sup>C2C/C2C</sup> IHCs than in *Otof* <sup>+/+</sup> IHCs (Kolmogorov-Smirnov test, p<10<sup>−3</sup>; *Figure 10D*), probably due to the defective vesicle fusion and vesicle pool replenishment in *Otof* <sup>C2C/C2C</sup> IHCs. Our results do not support the existence of a specific effect of the C<sub>2</sub>C domain mutations on one release mechanism — uniquantal (*Chapochnikov et al., 2014*) or multiquantal (*Li et al., 2014*) — rather than the other. Overall, our postsynaptic EPSC recordings confirmed that the synaptic vesicles of *Otof* <sup>C2C/C2C</sup> IHCs can fuse with the plasma membrane, but at a slower rate than those of *Otof* <sup>+/+</sup> IHCs.

## Modeling of IHC synaptic release supports a dual role for otoferlin in Ca<sup>2+</sup> sensing for vesicle fusion and vesicle pool replenishment

We used a previously described simplified model of IHC synaptic vesicle exocytosis (*Schnee et al., 2005*, *2011b*) to obtain a quantitative estimate of the in vivo Ca<sup>2+</sup>-sensing characteristics of otoferlin in RRP fusion and linear and superlinear release (*Figure 11A*). This model considers four vesicle pools (1 to 4) with different dynamics: pools 1, 2, and 3 correspond to the functionally defined RRP, recycling pool, and reserve pools, respectively. A 'distant pool' (DP, pool 4), equivalent to the entire

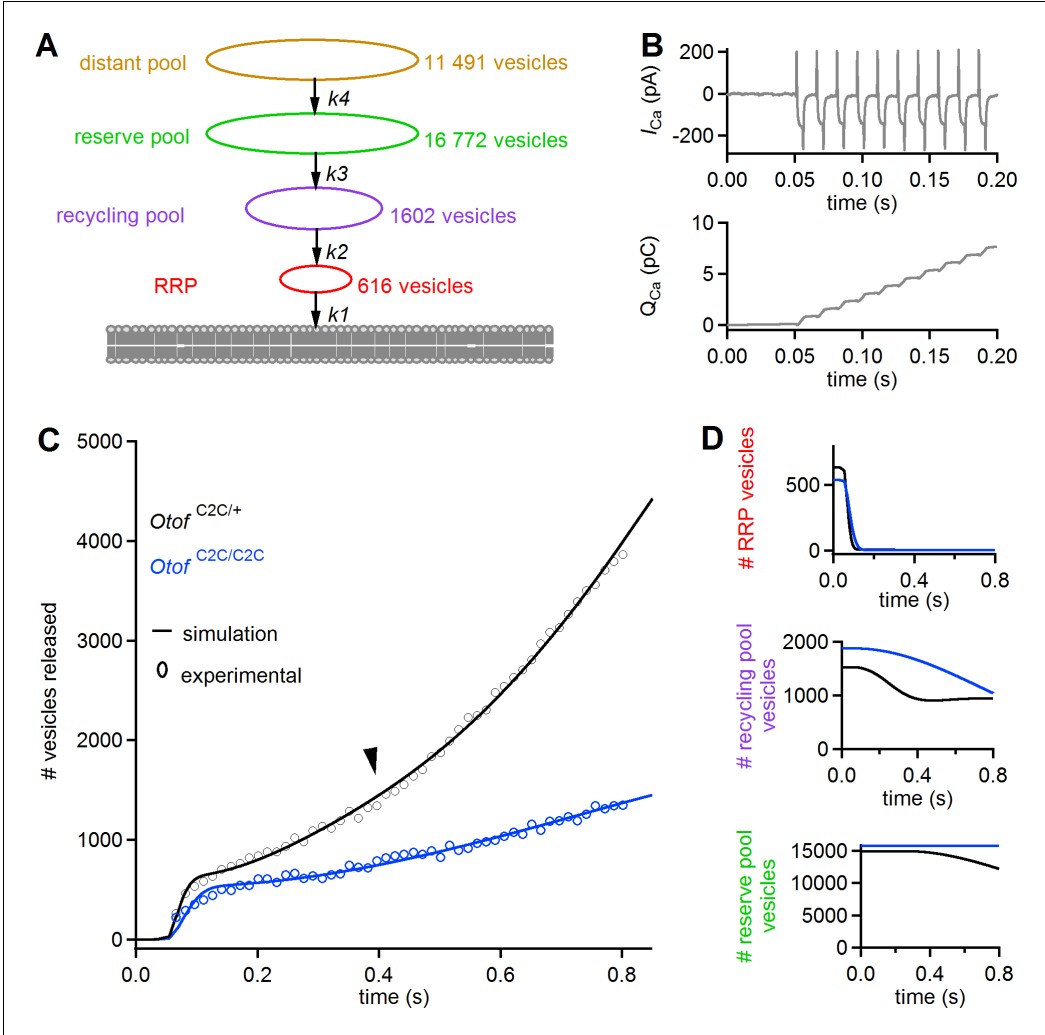

**Figure 11.** A mass action model of synaptic release reproduces Ca$^{2+}$ sensitivity defects in *Otof* $^{C2C/C2C}$ IHCs. (**A**) Diagram of the synaptic vesicle pools described by the model: the RRP (red), the recycling pool (violet), the reserve pool (green), and the distant pool (yellow). The process of synaptic exocytosis is governed by four rate constants, for RRP vesicle fusion ($k_1$), replenishment of the RRP from the recycling pool ($k_2$), replenishment of the recycling pool from the reserve pool ($k_3$), and replenishment of the reserve pool from the distant pool ($k_4$). The vesicle numbers indicated for each pool are the results obtained by least-squares fitting to the experimental data for *Otof* $^{C2C/+}$ IHCs. (**B**) Original Ca$^{2+}$ current ($I_{Ca}$) trace (recorded without the sine wave variation of the holding potential used to determine C$_m$) elicited by a train of 50 successive 5 ms depolarizations in an *Otof* $^{C2C/+}$ IHC (upper panel), and the corresponding integrated charge Q$_{Ca}$ as a function of time (lower panel). (**C**) Experimental mean ΔC$_m$ data as in *Figure 8F*, converted into the number of fused vesicles (black and blue circles) during a train of 50 successive 5 ms depolarizations, superimposed onto the best least-squares fits (black and blue lines) of the model for *Otof* $^{C2C/+}$ and *Otof* $^{C2C/C2C}$ IHCs (see *Table 1*). (**D**) Corresponding simulation of changes in vesicle numbers for each vesicle pool in *Otof* $^{C2C/+}$ and *Otof* $^{C2C/C2C}$ IHCs.

DOI: https://doi.org/10.7554/eLife.31013.020

IHC synaptic vesicle reservoir, is also included in the model. In normal conditions, this pool has little influence on the dynamics of the other pools. The main assumptions of the model are as follows: each of the four pools has a fixed size limit; vesicles exiting one pool immediately enter the next pool closer to the fusion site; and all vesicles entering the RRP eventually fuse. The kinetics of the various pools are governed by four mass-action equations describing the transitions of vesicles from pool 4 to pool 1 and their ensuing fusion (with transition rates $K_4$, $K_3$, $K_2$, and $K_1$, respectively (see *Equations (7)-(10)* in Materials and methods), and Ca$^{2+}$ thresholds for the recruitment of each

vesicle pool (see *Equations (1)-(4)*, in Materials and methods)). We took the observed $Ca^{2+}$ dependence of replenishment into account by modifying the model of Schnee *et al.* (*Schnee et al., 2011b*) such that all transition rates were explicitly dependent on intracellular $Ca^{2+}$ concentration (see *Equations (1)-(4)*, in Materials and methods).

The input $Ca^{2+}$ concentration used was the integral of the $Ca^{2+}$ currents (taking the effective volume in which $Ca^{2+}$ diffusion occurs as a unit volume) recorded in *Otof* [C2C/+] mice (*Equation (5)* in Materials and methods) during trains of 50 depolarizations of 5 ms each, separated by 10 ms (*Figure 11B*, top). The values of the various model parameters (*Table 1*) were determined by least-squares fitting of all model parameters to experimental data (see *Table 1* and, *Figure 11C–D*). Under the conditions described in *Table 1*, this modified model faithfully reproduced the three dynamic components observed in *Otof* [C2C/+] IHCs (RRP depletion, linear and superlinear release; *Figure 11C–D*). No $Ca^{2+}$ load terms other than those in the equations were required to reproduce the superlinear release component. Least-squares fits of the model to the experimental results for *Otof* [C2C/C2C] IHCs yielded values for the RRP fusion rate ($K_1$) and the rate of transition from the recycling pool to the RRP ($K_2$) lower than those in *Otof* [C2C/+] IHCs, by factors of 3.0 and 7.7, respectively (*Table 1*). The rate of transition from the reserve pool to the recycling pool ($K_3$) was also markedly lower, but with a large confidence interval (*Table 1*). The $Ca^{2+}$ thresholds for the recruitment of each vesicle pool were similar in *Otof* [C2C/+] and *Otof* [C2C/C2C] IHCs. The sizes of the four vesicle pools were also similar in *Otof* [C2C/+] IHCs and *Otof* [C2C/C2C] IHCs, consistent with the electron tomography results (*Figure 4*). Changes in the $K_1$, $K_2$, and $K_3$ transition rates were, therefore, sufficient to reproduce the main features of the impairment of synaptic exocytosis observed in *Otof* [C2C/C2C] IHCs (slower RRP depletion, lower rate of linear release, and absence of superlinear release) (*Figure 11C–D*).

## Discussion

By genetically modifying the $Ca^{2+}$-binding properties of the otoferlin $C_2C$ domain, we were able to investigate the hypothetical $Ca^{2+}$-sensor function(s) of the protein in the mature IHC synapse in vivo. By contrast to previously reported *Otof* mouse mutants (*Roux et al., 2006*; *Longo-Guess et al., 2007*; *Pangrsic et al., 2010*; *Strenzke et al., 2016*), most morphological and functional features of the mature IHC synapse, and the amount and subcellular distribution of otoferlin were unchanged in the mutant *Otof* [C2C/C2C] mice. The structure of the IHC ribbon synapse, the sizes and densities of the different vesicle pools, and the distance of RRP synaptic vesicles from the presynaptic plasma membrane were similar in *Otof* [C2C/+] and *Otof* [C2C/C2C] mice. Only the distance between RAP vesicles and the ribbon was slightly larger in *Otof* [C2C/C2C] mice. The pool size of vesicles able to fuse was unaffected in *Otof* [C2C/C2C] IHCs, as shown by the normal maximal $\Delta C_m$ evoked by $Ca^{2+}$ uncaging.

**Table 1.** Values of the parameters used in the mass action model of synaptic release

| Parameter | *Otof* [C2C/+] | *Otof* [C2C/C2C] |
|---|---|---|
| $V_{M1}$ | 616 ± 105 vesicles | 570 ± 81 vesicles |
| $V_{M2}$ | 1 602 ± 248 vesicles | 1 877 ± 487 vesicles |
| $V_{M3}$ | 16772 ± 8792 vesicles | 11 312 ± 3835 vesicles |
| $V_{M4}$ | 11 491 ± 2015 vesicles | 14 171 ± 3562 vesicles |
| $k_1$ | 1062 ± 583 $C^{-1}.s^{-1}$ | 349 ± 169 $C^{-1}.s^{-1}$ |
| $k_2$ | 6.4 ± 3.9 $C^{-1}.s^{-1}$ | 0.8 ± 0.4 $C^{-1}.s^{-1}$ |
| $k_3$ | 4.5 ± 9.9 $C^{-1}.s^{-1}$ | 0.95 ± 0.86 $C^{-1}.s^{-1}$ |
| $k_4$ | (6.6 ± 5.2) x $10^{-3}$ $C^{-1}.s^{-1}$ | (8.5 ± 3.6) x $10^{-3}$ $C^{-1}.s^{-1}$ |
| $[Ca^{2+}]_1$ | (1.7 ± 0.8) x $10^{-14}$ C | (2.1 ± 0.1) x $10^{-14}$ C |
| $[Ca^{2+}]_2$ | (8.3 ± 5.9) x $10^{-13}$ C | (6.7 ± 2.7) x $10^{-13}$ C |
| $[Ca^{2+}]_3$ | (8. 7 ± 2. 6) x $10^{-12}$ C | (10.2 ± 1.4) x $10^{-12}$ C |
| $[Ca^{2+}]_4$ | (9.7 ± 3.4) x $10^{-12}$ C | (9.7 ± 1.6) x $10^{-12}$ C |

DOI: https://doi.org/10.7554/eLife.31013.021

The amplitude of voltage-triggered $Ca^{2+}$ currents and the spatiotemporal pattern of intracellular $Ca^{2+}$ diffusion were also unaffected. Finally, the tight coupling between $Ca^{2+}$ channels and RRP vesicles was unmodified, with no observed change in RRP fusion kinetics in the presence of high intracellular EGTA concentrations. The necessary conditions for assessment of the specific effects of the $C_2C$ mutations on IHC synaptic exocytosis were therefore met.

Convergent lines of evidence indicate that mutations of the otoferlin $C_2C$ domain affected the $Ca^{2+}$ sensing domains involved in triggering the fusion of RRP vesicles at the IHC ribbon synapse. Through the various experimental protocols used to manipulate intracellular $Ca^{2+}$ levels near release sites during $Ca^{2+}$ channel activation, we found that these mutations almost halved the $Ca^{2+}$ sensitivity of the RRP sensor (i.e., decrease by a factor of 1.7 based on the modulation of depolarization levels, of 1.5 for variation of the duration of depolarization, or of 2.2 for the variation of extracellular $Ca^{2+}$ concentrations). The lower $Ca^{2+}$ sensitivity of vesicle fusion in $Otof^{C2C/C2C}$ IHCs was further established by rapid increases in intracellular $Ca^{2+}$ concentration upon photolysis of caged $Ca^{2+}$ that resulted in a delayed onset of exocytosis and a doubling of the time taken to reach the maximal rate of fusion. By simultaneously monitoring $\Delta C_m$ and intracellular $Ca^{2+}$ concentration, we were able to show that the $C_2C$ domain mutation, by reducing the $Ca^{2+}$ affinity of otoferlin, affected the $Ca^{2+}$ sensitivity of synaptic release but with no major effect on its cooperativity. Finally, the results of the simulation derived from the modified mass-action model (*Schnee et al., 2011b*) were consistent with a lower rate of RRP vesicle fusion at $Otof^{C2C/C2C}$ IHC synapses. Together, our results show that otoferlin functions as a $Ca^{2+}$ sensor for vesicle fusion with the plasma membrane at the IHC ribbon synapse, and implicate the $C_2C$ domain in the $Ca^{2+}$ dependence of RRP vesicle fusion rate.

$Otof^{C2C/C2C}$ IHCs displayed sustained exocytosis in response to trains of depolarizing pulses (*Figure 8D–F*), but the rate for the linear component of this sustained release was about a third that in $Otof^{C2C/+}$ IHCs, and the superlinear component of this release observed in $Otof^{C2C/+}$ IHCs could be elicited in only one of 17 $Otof^{C2C/C2C}$ IHCs. The finding of normal vesicle numbers, densities, and distributions in the RAP and OP, which are thought to underlie RRP replenishment, excluded the possibility of a smaller number of vesicles being the cause of the lower rate of sustained release in $Otof^{C2C/C2C}$ IHCs. This conclusion was also supported by the unaffected maximal release in $Ca^{2+}$ uncaging experiments in these mutants. The paired-pulse ratio (for different inter-pulse intervals) experiments, probing synaptic vesicle pool replenishment efficiency independently of fusion efficiency (*Moser and Beutner, 2000*), indicated that peak replenishment efficiency in $Otof^{C2C/C2C}$ IHCs was only half that in $Otof^{C2C/+}$ IHCs, ruling out the possibility of the replenishment defect being merely a consequence of the impairment of RRP fusion. Consistent with this result, modification of the parameters governing the kinetics of RRP fusion alone in the model did not reproduce the slower rate of sustained release observed experimentally in $Otof^{C2C/C2C}$ IHCs. The simulations were consistent with the experimental results only if both the transition rate for vesicle fusion ($K_1$) and that for synaptic vesicle pool replenishment ($K_2$) were decreased. The detrimental effect of $C_2C$ mutations on synaptic vesicle pool replenishment is unlikely to be caused by an impairment of endocytosis, because endocytosis rates and synaptic vesicle numbers were similar in $Otof^{C2C/+}$ and $Otof^{C2C/C2C}$ IHCs. Finally, the RRP replenishment defect in $Otof^{C2C/C2C}$ IHCs was partially rescued, in terms of total synaptic release, by making $Ca^{2+}$ available at high concentrations throughout the cytoplasm, as in $Ca^{2+}$ uncaging experiments, suggesting that synaptic vesicle reformation is not affected in $Otof^{C2C/C2C}$ IHCs. The similar patterns of $Ca^{2+}$ entry and diffusion in $Otof^{C2C/+}$ and $Otof^{C2C/C2C}$ IHCs excluded a lower local $Ca^{2+}$ concentration as the cause of the superlinear release suppression in $Otof^{C2C/C2C}$ IHCs, and provided additional evidence that the lack of superlinear sustained release is due to the abnormal $Ca^{2+}$ sensing properties of the mutated otoferlin. We therefore conclude that the synaptic vesicle pool replenishment defect in $Otof^{C2C/C2C}$ IHCs is independent of the RRP fusion defect, and results from the slower recruitment of vesicles to the release site caused by the lower $Ca^{2+}$ affinity of the mutated otoferlin. The molecular motor myosin VI, which has been shown to interact with otoferlin (*Roux et al., 2006*; *Heidrych et al., 2009*), might be involved in this recycling process.

What role does the otoferlin $C_2C$ domain play in IHC synaptic vesicle fusion and in synaptic vesicle pool replenishment? Otoferlin interacts with neuronal t-SNARE proteins through the $C_2A$, $C_2B$, $C_2C$, $C_2E$ and $C_2F$ domains, and with the $Ca_v1.3$ channel through the $C_2A$, $C_2B$, $C_2D$ and $C_2F$ domains, but not the $C_2C$ domain, in vitro (*Roux et al., 2006*; *Ramakrishnan et al., 2009*; *Johnson and Chapman, 2010*; *Hams et al., 2017*). However, the presence of neuronal SNARE

proteins in IHCs is under debate (*Nouvian et al., 2011*). The *Otof* C2C/C2C mutant IHCs still showed significant vesicle fusion and vesicle pool replenishment. Moreover, by contrast to the findings for *Otof* -/- (*Roux et al., 2006*), *pachanga* mutant (*Pangrsic et al., 2010*), and *Otof* I515T/I515T mutant mice (*Strenzke et al., 2016*), all vesicles in *Otof* C2C/C2C IHCs were able to fuse when exposed to high $Ca^{2+}$ loads, such as those released in $Ca^{2+}$ uncaging experiments, although the fusion process was both slower and delayed. These results suggest that the lower $Ca^{2+}$-binding affinity of otoferlin due to the $C_2C$ mutations can be partially overcome by high $Ca^{2+}$ concentrations. Consistent with this hypothesis, exocytosis levels in *Otof* C2C/+ and *Otof* C2C/C2C IHCs were similar in particular recording conditions. For 20 ms-long IHC depolarizations of small amplitudes (to between −65 mV and −30 mV), resulting in few open $Ca^{2+}$ channels but strong $Ca^{2+}$ influx-driving forces, the $Ca^{2+}$ dependence of $\Delta C_m$ was similar in *Otof* C2C/C2C and *Otof* C2C/+ IHCs. At these membrane potentials, the $Ca^{2+}$ sensor of the RRP vesicles closest to the few open $Ca^{2+}$ channels is readily saturated with $Ca^{2+}$, triggering vesicle fusion at similar levels in *Otof* C2C/C2C and *Otof* C2C/+ IHCs. Based on the proposed role of the $C_2A$ domain in Syt1 (*Shin et al., 2009*), we suggest that $Ca^{2+}$ binding to the $C_2C$ domain of otoferlin enhances the activity of other $C_2$ domains, resulting in an increase in the overall $Ca^{2+}$-sensitivity of release. Mutations of the otoferlin encoding gene are among the most frequent causes of inherited profound deafness in humans. Gene therapy projects based on the transfer of an otoferlin 'minigene', a method similar to that being developed for dysferlin defects (*Sinnreich et al., 2006*; *Lek et al., 2013*; *Fuson et al., 2014*), require to extend the present functional characterization of the otoferlin $C_2C$ domain to the other $C_2$ domains of the protein.

What consequences do the $C_2C$ domain mutations have for hearing? In one-month-old *Otof* C2C/C2C mice, hearing thresholds were only slightly higher than normal, but the amplitude of the ABR wave-I, reflecting the synchronous activity of the auditory nerve fibers in vivo, was a third that in *Otof* C2C/+ mice. The lower amplitude of ABR wave-I in *Otof* C2C/C2C mice is probably due to significantly lower rates of synaptic release and/or a desynchronization of synaptic vesicle fusion. Accordingly, although the mean amplitudes and decay time constants of primary auditory neuron EPSCs were unchanged, the EPSC rate following the extracellular application of 25 mM KCl was smaller in *Otof* C2C/C2C mice than in *Otof* +/+ mice. This effect may be minimal on hearing in vivo since the receptor potentials of IHCs are unlikely to reach values beyond −20 mV upon sound stimulation (*Palmer and Russell, 1986*), that is they are expected to stay in a range at which not all $Ca^{2+}$ channels are open but the $Ca^{2+}$ driving force is relatively large, potentially leading to the local saturation of $Ca^{2+}$ sensors. In contrast, the much slower sustained release related to vesicle pool replenishment failure may affect hearing much more by contributing to a desynchronization of the auditory nerve fiber responses. Remarkably, despite the smaller ABR wave-I, the timing and amplitude of wave-II, which reflects the synchronous activity of the central auditory synaptic relays in the cochlear nucleus, were preserved in *Otof* C2C/C2C mice. It has been suggested that the anatomical convergence of many auditory nerve fibers onto principal neurons of the ventral cochlear nucleus may compensate to a certain degree the asynchronism effect of impaired IHC exocytosis on the response of auditory neurons (*Buran et al., 2010*).

We show here that the same $Ca^{2+}$ sensor, otoferlin, is involved in synaptic vesicle fusion and in synaptic vesicle pool replenishment at the IHC ribbon synapse. In many synapses, different steps of the synaptic vesicle cycle involve different $Ca^{2+}$ sensors. For instance, at the calyx of Held synapse, Syt2 drives synchronous vesicular exocytosis (*Sun et al., 2007*), and vesicle pool replenishment requires the $Ca^{2+}$-calmodulin-Munc13-1 complex (*Lipstein et al., 2013*). Likewise, rapid vesicular exocytosis at hippocampal synapses depends on Syt1, whereas vesicle pool replenishment involves Syt7 and calmodulin (*Liu et al., 2014*; *Jackman et al., 2016*). In some synapses, however, Syt1 or Syt2 are involved in several steps of the synaptic vesicle cycle. At the neuromuscular junction in drosophila, Syt1 is involved both in exocytosis and endocytosis (*Poskanzer et al., 2003*), and in mouse cerebellum basket cells, the major $Ca^{2+}$ sensor for exocytosis, Syt2, also mediates fast vesicle pool replenishment (*Chen et al., 2017*). In IHCs we cannot exclude the possibility of otoferlin being assisted by other $Ca^{2+}$-sensing proteins, such as Syt4, which has been shown to be involved in the developmental transition of exocytosis from nonlinear to linear $Ca^{2+}$ dependence (*Johnson et al., 2010*). Mature IHC ribbon synapses lack several proteins critical for vesicle fusion in synapses of the central nervous system, including Syt1 and Syt2 (*Safieddine and Wenthold, 1999*; *Beurg et al., 2010*), complexins, which act as important regulators of spontaneous and fast synchronous $Ca^{2+}$-evoked fusion (*Giraudo et al., 2006*; *Strenzke et al., 2009*; *Krishnakumar et al., 2011*; *Lai et al.,*

2014), synaptophysin (*Safieddine and Wenthold, 1999*), and Munc13 (*Vogl et al., 2015*). In addition, neuronal SNARE proteins (synaptobrevins, snap-25, and syntaxin-1) have been reported to be dispensable for exocytosis at the IHC ribbon synapse, suggesting that other molecular components take on the role of these proteins (*Nouvian et al., 2011*). Such a singular molecular setting of the mature IHC ribbon synapse presumably optimizes the coordination between fast vesicle fusion and synaptic vesicle pool replenishment, allowing this synapse to operate indefatigably at high rates and with a high degree of temporal precision.

## Materials and methods

### Animals

Animal experiments were carried out in accordance with European Community Council Directive 2010/63/UE under authorizations 2012–028, 2012–038, and 2014–005 from the Institut Pasteur ethics committee for animal experimentation. The $Otof^{Ala515,Ala517/Ala515,Ala517}$ (referred to as $Otof^{C2C/C2C}$) knock-in mouse mutant was generated by homologous recombination (Institut Clinique de la Souris, Illkirch, France). The targeting construct, spanning exon 15 (ENSEMBL ENSMUSE00001209343) and containing the two missense mutations, was introduced by electroporation into embryonic stem cells from the 129S1/SvlmJ mouse strain. Stem cells carrying the desired construct were injected into blastocysts from C57BL/6J mice to obtain chimeric mice. After germline transmission, mice were crossed with C57BL/6J mice. Birth rates for all genotypes conformed to Mendelian ratios. Experiments were carried out on both male and female mice, mostly obtained by crossing heterozygous mice ($Otof^{C2C/+}$) with homozygous mutant mice ($Otof^{C2C/C2C}$), genotyped by PCR with the following primers: forward 5'-ATTACCTCTGCTGCTTTTGCACCTG-3' and reverse 5'-CTCAGCAGGTGCTTCTGACCAC-3', spanning the excised selection marker region in intron 15. As IHC synaptic exocytosis was similar in $Otof^{C2C/+}$ and $Otof^{+/+}$ (i.e. wild-type) mice, this breeding scheme was used to produce substantial numbers of $Otof^{C2C/C2C}$ homozygous mutant mice and $Otof^{C2C/+}$ mice (used as controls) within a given mouse litter. For all experiences, except otoferlin immunofluorescence quantification, experimentalists were not blinded against the genotypes of the mice.

### Immunohistofluorescence

The entire mouse inner ear was isolated from the surrounding bone, as previously described (*Roux et al., 2006*). For rapid fixation of the cochlear sensory epithelium (organ of Corti), the round and oval windows were opened, the bone over the cochlear apical turn was removed, and the cochlea was perfused with 4% paraformaldehyde in phosphate-buffered saline (PBS). The cochlea was then post-fixed by incubation in the same fixative for 30 min at 4°C. Whenever the anti-Ca$_v$1.3 antibody was used, the cochlea was fixed by incubation with 99% methanol for 20 min at −20°C. Cochlear whole-mount preparations were permeabilized by incubation with 0.3% Triton X-100 in PBS supplemented with 20% normal horse serum for one hour at room temperature. 4',6-diamidino-2-phenylindole (DAPI) was used to stain cell nuclei. The following antibodies were used: rabbit anti-otoferlin (1:250 dilution) (*Roux et al., 2006*), goat anti-CtBP2/ribeye (1:150 dilution; Santa Cruz Biotechnology; # sc-5966), rabbit anti-Ca$_v$1.3 (1:50 dilution; Alomone Labs; #ACC-005), mouse anti-GluA2 (1:200 dilution; Millipore; #MAB397), rabbit anti-Vglut3 (1:250, Synaptic Systems, # 135 203) and secondary Atto Fluor Cy5-conjugated anti-mouse, Alexa Fluor 488-conjugated anti-goat, and Atto Fluor 647-conjugated anti-rabbit IgG (1:200 dilution, Sigma) antibodies. Anti-Ctbp2, anti-GluA2 and anti-Ca$_v$1.3 antibodies have been widely used and shown to be specific (*Beurg et al., 2010*; *Liberman et al., 2011*; *Jing et al., 2013*; *Wong et al., 2014*; *Vincent et al., 2017*). After incubation with the appropriate antibodies, the samples were washed three times in PBS, and mounted in one drop of Fluorsave medium (Biochem Laboratories, France). Fluorescence confocal *z*-stacks from selected cochlear regions were obtained with an LSM 700 confocal microscope (Zeiss, Oberkochen, Germany) equipped with a high-resolution objective (numerical aperture of 1.4, 60 x oil-immersion objective). Images were acquired with a 4 x digital zoom in a 1024 × 1024 raster (pixel size = 0.036 μm in *x* and *y*) with 0.2 μm steps in *z*. Three $Otof^{C2C/+}$ and five $Otof^{C2C/C2C}$ mice were examined for the analysis of otoferlin labeling. Four $Otof^{C2C/+}$ and four $Otof^{C2C/C2C}$ mice were examined for determination of the number of ribbon synapses per IHC.

For otoferlin immunofluorescence quantification, whole-mount preparations of organs of Corti from *Otof* [C2C/+] and *Otof* [C2C/C2C] littermates were blind-processed in parallel, using the same experimental settings from tissue fixation to confocal imaging. For each organ of Corti, fluorescence intensity was averaged in Gaussian volumes with standard deviations of 1 µm along the X, Y and Z axes, centered around points selected at the apex, the middle, and the base of 7–8 IHCs per confocal stack (for a total of 51 *Otof* [C2C/+] IHCs and 52 *Otof* [C2C/C2C] IHCs). In effect, the 3D images were blurred by convolution with a Gaussian of the prescribed standard deviations, and intensity values of the blurred images at the selected points were retrieved.

## STED microscopy

Dual-color STED microscopy was carried out with a custom-built system (*Lauterbach et al., 2013*) using two excitation beams at 480 ± 10 nm and 532 ± 5 nm obtained by filtering a super-continuum laser beam, and one STED beam tuned at 647 nm, coupled with a helical phase mask (VPP-A1, RPC Photonics, New York) to produce a toroidal (donut-shaped) diffraction pattern centered on the excitation spot in the focal plane of a 100x/1.4NA objective lens (Olympus, Tokyo, Japan). Since the two excitation beams arise from a single monomode optical fiber, the two color-channels are co-aligned by design. Two-color STED imaging was achieved with two dyes, Atto565 and DY485XL, excited with the 532 nm and 480 nm excitation beams, respectively. Atto565 has excitation and emission peaks around 563 nm and 592 nm, respectively, whereas the long Stokes-shift dye DY485XL displays maximal excitation at about 485 nm, with an emission peak at around 560 nm. Fluorescence images were sequentially collected with an avalanche photodiode (Perkin Elmer) behind a 585/65 emission filter. A pixel size of 50 nm and a scanning dwell time of 100 µs were used for the acquisitions.

## STED microscopy image processing and colocalization analysis

Wavelet denoising and deconvolution were applied to the images, as previously described, with a point spread function extracted from the images (*de Monvel et al., 2003*), and linear unmixing was used to compensate for slight leakage between the two excitation channels. The staining patterns observed in the Atto565- and the DY485-channel images was granular, and consisted in many resolution-limited spots representing structures stained for otoferlin and Vglut3, referred to as otoferlin spots and Vglut3 spots, respectively, within the cells. The densities of either the Vglut3 or the otoferlin spots were similar within all the cells analyzed (p>0.05, Kolmogorov-Smirnov test). This made it possible to analyze the colocalization of otoferlin and Vglut3 spots by determination of the positions of each type of spots within the cells of interest, which could be achieved with a resolution better than the pixel size by maxima detection. Nearest-neighbor analysis was then performed by comparing the distribution of the distance between a randomly chosen otoferlin spot and the nearest Vglut3 spot, to quantify the colocalization of the two proteins, with the distribution of the distance between a random Vglut3 spot and the nearest neighboring Vglut3 spot, corresponding to the expected distribution of nearest-neighbor distances for randomly distributed points. The shift of the otoferlin-Vglut3 nearest-neighbor distance distribution towards distance values smaller than expected for a purely random distribution indicated colocalization of the two proteins.

## Electron tomography

Cochleas were perfused with 4% paraformaldehyde and 2% glutaraldehyde in Sorensen buffer at pH 7.4, and immersed in the fixative solution for 2 hr. They were then postfixed by incubation overnight in 1% osmium tetraoxide in cacodylate buffer at 4°C. They were dehydrated in a graded series of acetone concentrations and embedded in Spurr's low-viscosity epoxy resin (EMS, Hatfield, USA), which was then hardened at 70°C. For tomographic analysis, thick (200 or 250 nm) sections of the sensory epithelium (organ of Corti) were collected on 100-mesh parallel bar copper grids, and incubated, for 10 min on each side, with 15 nm protein gold particles (UMC Utrecht, The Netherlands). The grids were contrast-stained with 4% uranyl acetate in dH$_2$O for 40 min, followed by Reynold's lead citrate for 3 min. The sections were viewed in an FEI *Tecnai G2 200kV* transmission electron microscope, and single- or double-tilt series were acquired from approximately −65° to +65° with 1° increments, with FEI Xplore 3D software and a Gatan US 4000 camera. The acquired tilt series were processed with a wavelet-denoising algorithm implemented in Matlab (Mathworks) (*Boutet de Monvel et al., 2001*) to reduce background noise without losing fine detail. The images of the

tomographic tilt series were then aligned, and the final volume was reconstructed with a weighted back-projection algorithm and IMOD software (*Kremer et al., 1996*).

## 3D reconstructions and estimates of vesicle pool sizes

Analyses of the ribbon synapses, including segmentation, 3D reconstruction, and rendering, were carried out with AMIRA software (version 5.1; Mercury Computer Systems, San Diego, CA) and with custom Matlab functions (Mathworks). The contours of the ribbon, the presynaptic density of the afferent dendrite, and nearby organelles, such as mitochondria, coated pits, and tubular structures, were drawn on every section. Spheres of constant diameter were used to mark synaptic vesicles. The ribbon was defined as the center of the active zone. For each ribbon, we counted the number of synaptic vesicles within 80 nm of the ribbon surface. These vesicles were considered to constitute the ribbon-attached vesicle pool (RAP), which is thought to correspond to the functional recycling and reserve pools (*Rizzoli and Betz, 2005*). A subset of the ribbon-attached vesicles, with centers lying within 40 nm of the presynaptic membrane and below the ribbon (within 80 nm of the center of the active zone), was considered to form the pool of docked or readily releasable vesicles (the RRP), thought to be released first during depolarization (*Lenzi et al., 1999*; *Schnee et al., 2011a*). We chose this distance of 40 nm, because the mean radius of a vesicle was ~20 nm and because the cytosolic parts of t-SNARE and v-SNARE are ~10 nm long, so SNARE interactions may occur at distances of up to 20 nm from the presynaptic plasma membrane (*Zenisek et al., 2000*; *Castorph et al., 2010*). Using our ribbon reconstruction data and taking the distribution of synaptic vesicles into account, we estimated the total size and volume density of the synaptic vesicles attached to each ribbon, and the number and volume density of outlying cytoplasmic vesicles located within 350 nm of the ribbon surface — the outlying vesicle pool (OP) thought to contribute to the functional reserve pool (*Rizzoli and Betz, 2005*). We estimated these vesicle pools in our tomographic reconstructions of ribbon synapses, using only ribbon reconstructions including more than half of the ribbon surface. The numbers and densities of vesicles in each pool were obtained with a distance transformation (implemented in Matlab) providing volume shells around the ribbon delimited by various distances from the ribbon surface.

## Functional hearing tests

Auditory brainstem responses (ABRs) and distortion product otoacoustic emissions (DPOAEs) were recorded, as previously described (*Le Calvez et al., 1998*), in mice aged between 1 and 13 months. ABR waves were recorded in response to pure tone bursts at sound frequencies of 10, 15, 20, and 32 kHz. ABR signals were averaged after the presentation of a series of 100–200 pure tone bursts. ABR thresholds were defined as the lowest stimulus for which recognizable ABR waves could be observed. The amplitude of ABR wave-I was estimated by measuring the voltage difference between the wave-I peak and the trough between wave-I and wave-II.

DPOAEs were collected with a miniature microphone positioned at the entry of the ear canal. Two primary pure-tone stimuli of frequencies $f_1$ and $f_2$ were applied simultaneously, with $f_2$ between 5 and 20 kHz, and the $f_2/f_1$ ratio kept constant at 1.2. The cubic difference tone at $2f_1 - f_2$, the most prominent distortion product tone produced by the mammalian ear, was measured for primary tone frequencies of equal levels, ranging from 30 to 75 dB SPL.

## Electrophysiology and Ca²⁺ imaging

Electrophysiological recordings were carried out on excised cochlear apical coils of either prehearing (P8-P12) or hearing (P15-P18) $Otof^{+/+}$, $Otof^{C2C/+}$, and $Otof^{C2C/C2C}$ mice, as previously described (*Beurg et al., 2010*). The dissection solution contained 143 mM NaCl, 6 mM KCl, 1.3 mM CaCl$_2$, 0.9 mM MgCl$_2$, 0.7 mM NaH$_2$PO$_4$, 5 mM glucose, 2 mM sodium pyruvate, 10 mM HEPES, pH 7.4. Recordings were carried out at room temperature (20–23°C). The patch pipette electrodes were made of borosilicate glass (World Precision Instruments).

For cell capacitance experiments on IHCs, the pipette resistance in the external solution was 2–3 MΩ. Only cells with a series resistance below 10 MΩ (uncompensated) were included in the study. Ca²⁺ current and $\Delta C_m$ were recorded with an EPC-9 patch-clamp amplifier and Patchmaster software (HEKA, Ludwigshafen, Germany). We used a single 30 mV amplitude sine wave from a holding potential of −95 mV. Except for endocytosis recordings, the acquisition frequency was 50 kHz, low-

pass filtered at 6–10 kHz, and the frequency of the sine wave was 1 kHz. In the case of endocytosis recordings, the acquisition frequency was set to 10 kHz, and the frequency of the sine wave to 800 Hz. The resulting maximal depolarization to ~−65 mV was sufficiently small to avoid activation of the $Ca^{2+}$ current. $Ca^{2+}$ current recordings were corrected for the linear leak conductance measured near −95 mV. Liquid junction potential (~−15.5 mV) was corrected off-line for Cs-gluconate-based intracellular solutions. The extracellular recording solution consisted of 111.5 mM NaCl, 6 mM KCl, 10 mM $CaCl_2$, 1 mM $MgCl_2$, 27 mM TEA-Cl, 2 mM sodium pyruvate, 5 mM glucose, 10 mM Na-HEPES, pH 7.4 (adjusted with NaOH). Tetrodotoxin (1 µM) and apamin (1 µM) were added to the extracellular solution. The intracellular pipette solution for cell membrane capacitance recordings contained 140 mM Cs-gluconate, 20 mM TEA-Cl, 0.5 mM EGTA, 5 mM creatine phosphate, 4 mM Mg-ATP, 0.3 mM $Na_2$-GTP, 10 mM HEPES, pH 7.2 (adjusted with CsOH). We used 10 mM extracellular $Ca^{2+}$ and 0.5 mM intracellular EGTA unless otherwise specified. The $\Delta C_m$ evoked by membrane depolarization was measured as $\Delta C_m = C_{m\ (response)} - C_{m\ (baseline)}$, and was used as a measure of synaptic vesicle exocytosis in IHCs. $C_{m\ (baseline)}$ was obtained by averaging capacitance data points before the depolarizing pulse, and $C_{m\ (response)}$ was obtained by averaging capacitance data points after the transient current following the depolarizing pulse (*Figure 5—figure supplement 1A*). With depolarizing protocols, possible kinetic perturbations in $Otof^{C2C/C2C}$ IHCs are likely to be masked by the relatively long time interval between the end of the depolarizing stimulus and the measurement of the post-stimulus $C_m$ (typically 50–100 ms for a depolarization lasting 20 ms).

$Ca^{2+}$ imaging experiments were carried out by adding 200 µM Oregon Green Bapta 5N dye (OGB-5N, Invitrogen) to the intracellular patch pipette solution. Images were acquired at a rate of 500 Hz, with a high-speed camera (Photometrics) mounted on a swept-field confocal microscope (Bruker, Billerica, MA, USA). Imaging protocols began two minutes after cell opening, to allow the OGB-5N dye to diffuse. $Ca^{2+}$ entry was assessed by selecting regions of interest (a circle with a 6-pixel radius), in which the standard deviation of the fluorescent $Ca^{2+}$ signal was maximal during IHC depolarization (see the images in *Figure 8—figure supplement 1A*).

## $Ca^{2+}$ uncaging experiments

We triggered a rapid rise in the intracellular $Ca^{2+}$ concentration from the $Ca^{2+}$ caged by the DM-nitrophen chelator (Interchim, France), using a single flash from a high-power UV LED light source (Mic-LED 365, 128 mW, Prizmatix, Givat Shmuel, Israel). The UV LED TTL triggered through the EPC10 patch clamp amplifier, was directly connected to the epi-illumination port at the rear of an upright Nikon FN1 microscope, and illumination was focused through the 60 x objective (CFI Fluor 60 x W NIR, WD = 2.0 mm, NA = 1). Hair cells were loaded with 145 mM CsCl, 5 mM HEPES, 20 mM TEA, 10 mM DM-nitrophen and 10 mM $CaCl_2$. After patch rupture, we systematically kept the cells at a holding potential of −70 mV (without leak correction) for 2 min, for cell loading and equilibration.

In some experiments, intracellular $Ca^{2+}$ concentration was measured by adding 50 µM OGB-5N to the intracellular solution, as previously described (*Vincent et al., 2014*; *Vincent et al., 2015*). Changes in $[Ca^{2+}]_i$ were measured with a C2 confocal system and NIS-elements imaging software (Nikon, Japan) coupled to the FN1 Nikon upright microscope. The dye was excited with a 488 nm solid-state laser (85-BCD-010–706, Melles Griot, CA USA) and emission was recorded at 500–530 nm. OGB-5N-emission fluorescence was continuously monitored before and after the UV flash, by fast line scan analysis at 1 kHz spanning the middle part of the IHC. The $Ca^{2+}$-dependent fluorescent signal of OGB-5N was calibrated in situ, in the hair cell cytoplasm, as previously described (*Vincent et al., 2014*). Hair cells were voltage-clamped at −70 mV in the whole-cell configuration with different solutions containing various free calcium concentrations ($[Ca^{2+}]_{free}$), from 1 to 100 µM. The patch pipettes were filled with a Cs-based solution (145 mM CsCl, 5 mM HEPES, 20 mM TEA, 0.05 mM OGB-5N 0.05) and various concentrations of $Ca^{2+}$ and EGTA. The Ca-EGTA Maxchelator freeware (Stanford, CA, USA) was used to determine $[Ca^{2+}]_{free}$. After 2 min of cell loading and equilibrium, the fluorescence signal was measured at each $[Ca^{2+}]_{free}$ in different cells. Fluorescence emission values were plotted as $F/F_{max}$ against $[Ca^{2+}]_{free}$. Data points were best fitted by a sigmoidal function with a $K_D$ of 23.30 ± 0.71 µM. This $K_D$ value is very close to the value reported in vitro ($K_D$ = 20 µM, Molecular Probes, Life Technologies). We quantified $Ca^{2+}$ responses in each hair cell during the $Ca^{2+}$ uncaging experiments, by calculating the intracellular $Ca^{2+}$ concentration ($[Ca^{2+}]_i$) as follows (*[Pawley, 2006]*, chapter 42): $[Ca^{2+}]_i = K_D\ (F-F_{min})/(F_{max}-F)$, where $K_D$ is the estimated

constant of dissociation, F the fluorescence at time $t$, $F_{min}$ the minimum fluorescence, and $F_{max}$ the maximal fluorescence.

$C_m$ recordings in $Ca^{2+}$ uncaging experiments were performed with an EPC-10 patch-clamp amplifier. The amplitude and frequency of the sine wave were 20 mV and 1 kHz, respectively.

## EPSC recordings

Postsynaptic bouton recordings of EPSCs were performed as previously described (*Glowatzki and Fuchs, 2002*). Pipette resistance in the external solution was 8–10 MΩ. The external solution contained 5.8 mM KCl, 144 mM NaCl, 0.9 mM $MgCl_2$, 1.3 mM $CaCl_2$, 0.7 mM $NaH_2PO_4$, 5.6 mM glucose, 10 mM HEPES, pH 7.4 (adjusted with NaOH), and the pipette solution for intracellular recording contained 135 mM KCl, 3.5 mM $MgCl_2$, 0.1 mM $CaCl_2$, 5 mM EGTA, 5 mM HEPES, 2.5 mM $Na_2ATP$, pH 7.2 (adjusted with KOH). EPSCs were induced by increasing extracellular $K^+$ concentration from 5.8 mM to 25.8 mM, by replacing 20 mM NaCl with 20 mM KCl in the external solution. EPSC recordings were performed with a MultiClamp 700B amplifier (Molecular Devices), and a National Instruments digitizer (NI-PCIe 6351). They were low-pass filtered at 6 kHz, and digitized at 50 kHz.

Data were analyzed as previously described (*Glowatzki and Fuchs, 2002*; *Goutman and Glowatzki, 2007*). In the case of overlapping EPSCs, the amplitude of the second EPSC was estimated by fitting the decay of the first EPSC and subtracting the fitted value at the time of the second peak. Decay time constants were calculated with monophasic recorded EPSCs only.

## Statistical analysis

The data were analyzed with Igor Pro (WaveMetrics, Portland, OR, USA) and Prism (Graphpad, La Jolla, CA, USA) softwares. Values of $p<0.05$ were considered to indicate that the differences observed between groups were statistically significant. Two-tailed unpaired Student's $t$ tests with Welch's correction, which does not assume equal variances, were used, unless otherwise stated. The other tests performed included two-way ANOVA with Bonferroni post hoc tests to assess the interaction between two independent variables, Kolmogorov-Smirnov tests to compare distributions, Fisher's exact tests to compare proportions, and nonparametric Mann-Whitney tests when the data could not be assumed to be normally distributed, precluding use of the parametric Student's $t$ test. The normality of data distribution was assessed with the D'Agostino and Pearson normality test. Data are expressed as the mean ± standard error of the mean (SEM) unless otherwise stated. Numbers ($n$) in the figures and text indicate the number of biological replicates derived from independent experiments. Asterisks on bar graphs denote the statistical significance of the differences indicated in brackets ($^*p<0.05$; $^{**}p<0.01$; and $^{***}p<0.001$), whereas ns indicates 'not significant' ($p>0.05$).

## Modeling

Simulations of synaptic vesicle fusion in $Otof\ ^{C2C/C2C}$ and $Otof\ ^{C2C/+}$ IHCs were carried out with a modified version of a published mass action model of synaptic vesicle fusion in IHCs (*Schnee et al., 2005*; *Schnee et al., 2011b*) (*Figure 11A*). This model considers four functional vesicle pools: the RRP, the recycling pool, the reserve pool (*Rizzoli and Betz, 2005*), and a distant pool equivalent to the entire reservoir of IHC vesicles (*Schnee et al., 2011b*). Vesicle trafficking between pools is governed by five first-order differential equations. The transition rate constants for RRP vesicle fusion ($K_1$), and for vesicle trafficking from the recycling pool to the RRP ($K_2$), from the reserve pool to the recycling pool ($K_3$), and from the distant pool to the reserve pool ($K_4$) (see *Figure 11A*), were set so as to depend on intracellular $Ca^{2+}$ concentration ($[Ca^{2+}]_i$) (here expressed in units of charge (C), which corresponds to taking the effective volume in which $Ca^{2+}$ diffusion occurs as a unit volume) as follows:

$$K_1(t) = k_1 \cdot \max([Ca^{2+}]_i(t) - [Ca^{2+}]_1; 0) \tag{1}$$

$$K_2(t) = k_2 \cdot \max([Ca^{2+}]_i(t) - [Ca^{2+}]_2; 0) \tag{2}$$

$$K_3(t) = k_3 \cdot \max([Ca^{2+}]_i(t) - [Ca^{2+}]_3; 0) \tag{3}$$

$$K_4(t) = k_4 \cdot \max([Ca^{2+}]_i(t) - [Ca^{2+}]_4; 0) \tag{4}$$

$$[Ca^{2+}]_i(t) = \int^t I_{Ca}(t).dt \tag{5}$$

where $k_1$, $k_2$, $k_3$, and $k_4$ are $Ca^{2+}$-independent rate constants associated with the various pools, $[Ca^{2+}]_1$, …, $[Ca^{2+}]_4$ represent the minimum $Ca^{2+}$ concentrations triggering RRP fusion, and vesicle transitions from the recycling pool, the reserve pool, and the distant pool, respectively (*Figure 11A*). Minimum $Ca^{2+}$ concentrations were set to mimic recruitment of the additional pools of vesicles with the delays observed experimentally. Vesicle trafficking between pools and fusion were also constrained by the use of a maximum vesicle number: $V_{M1}$, $V_{M2}$, $V_{M3}$, and $V_{M4}$ for the RRP, recycling pool, reserve pool, and distant pool, respectively. These properties give rise to the following five equations governing the number of fused vesicles F(t) and vesicle number V(t) in each of the four vesicle pools:

$$dF(t)/dt = K_1(t) \cdot V_1(t) \tag{6}$$

$$dV_1(t)/dt = K_2(t) \cdot V_2(t) \cdot ((V_{M1} - V_1(t))/V_{M1}) - K_1(t) \cdot V_1(t) \tag{7}$$

$$dV_2(t)/dt = K_3(t) \cdot V_3(t) \cdot ((V_{M2} - V_2(t))/V_{M2}) - K_2(t) \cdot V_2(t) \cdot ((V_{M1} - V_1(t))/V_{M1}) \tag{8}$$

$$dV_3(t)/dt = K_4(t) \cdot V_4(t) \cdot ((V_{M3} - V_3(t))/V_{M3}) - K_3(t) \cdot V_3(t) \cdot ((V_{M2} - V_2(t))/V_{M2}) \tag{9}$$

$$dV_4(t)/dt = -K_4(t) \cdot V_4(t) \cdot ((V_{M3} - V_3(t))/V_{M3}) \tag{10}$$

Initial conditions were set as follows:

$$F(t=0) = 0, V_i(t=0) = V_{Mi}, i = 1, ..., 4. \tag{11}$$

Equations were implemented in Igor Pro 6 software (Wavemetrics) and in Matlab. The parameters $[Ca^{2+}]_1$, $[Ca^{2+}]_2$, $[Ca^{2+}]_3$, $[Ca^{2+}]_4$, $k_1$, $k_2$, $k_3$, $k_4$, $V_{M1}$, $V_{M2}$, $V_{M3}$, and $V_{M4}$ were obtained by least-squares fitting to the experimental data obtained for *Otof* [C2C/+] and *Otof* [C2C/C2C] IHCs. In detail, the sum-of-squares error on the $C_m$ curve was defined as the sum of squared differences between the averaged $C_m$ values measured as a function of some experimental parameter (depolarisation amplitude, duration of stimulus, or number of depolarisation pulses) and the corresponding $C_m$ values predicted by the model for a given set of fitting parameters. This error was minimized using the Matlab built-in minimization function (fminsearch function), which is a general-purpose nonlinear minimization function based on the simplex algorithm (*Lagarias et al., 1998*). Weighting of the squared errors was uniform. The following interval constraints were applied to the fitting parameters at all iterations: $k_1$,…, $k_4$ between 0 and $10^4$; $V_{M1}$, $V_{M2}$ between 0 and $10^5$; $V_{M3}$ between 5000 and $10^5$, and $V_{M4}$ between 10000 and $10^5$; $[Ca^{2+}]_1$,…, $[Ca^{2+}]_4$ between 0 and 10. Error estimates on the fitted parameters were obtained by a Monte-Carlo sensitivity analysis of the fit. Namely, we generated 60 randomized versions of the $C_m$ curves, assuming for each data point a gaussian distribution with the same mean and standard deviation as observed experimentally; we applied the same fitting procedure on each randomized $C_m$ curve. The mean values and standard deviations stated in *Table 1* were computed from 60 such Monte Carlo runs performed for each phenotype.

## Acknowledgements

We thank Dominique Weil and the *Institut Clinique de la Souris* (Illkirch, France) for providing us with the recombinant *Otof* [C2C/C2C] mice, and Philippe Vincent for advice for the $Ca^{2+}$ uncaging experiments. This work was supported by *Foundation Raymonde et Guy Strittmatter*, *Foundation BNP Paribas* and *LHW-Stiftung* grants to CP. This work was carried out in the framework of the LabEx Lifesenses [ANR-10-LABX-65] and was supported by French state funds managed by the *Agence*

Nationale pour la Recherche within the Investissements d'Avenir program under reference ANR-11-IDEX-0004-02, and the Prix Emergence of the Agir pour l'Audition foundation to NM.

## Additional information

### Competing interests
Christine Petit: Reviewing editor, *eLife*. The other authors declare that no competing interests exist.

### Funding

| Funder | Grant reference number | Author |
|---|---|---|
| Foundation Raymonde et Guy Strittmatter | Research project grant | Christine Petit |
| Fondation BNP Paribas | Research project grant | Christine Petit |
| LHW-Stiftung | Research project grant | Christine Petit |
| LabExLifesenses | ANR-10-LABX-65 | Christine Petit |
| Investissements d'Avenir | ANR-11-IDEX-0004-02 | Christine Petit |
| Agir pour l'Audition | Prix Emergence scientifique | Nicolas Michalski |

The funders had no role in study design, data collection and interpretation, or the decision to submit the work for publication.

### Author contributions
Nicolas Michalski, Conceptualization, Formal analysis, Supervision, Validation, Investigation, Visualization, Methodology, Writing—original draft, Project administration, Writing—review and editing; Juan D Goutman, Jacques Boutet de Monvel, Roger Bryan Sutton, Paul Avan, Formal analysis, Validation, Investigation, Visualization, Methodology, Writing—review and editing; Sarah Marie Auclair, Danica Ciric, Formal analysis, Investigation; Margot Tertrais, Alice Emptoz, Alexandre Parrin, Sylvie Nouaille, Investigation; Marc Guillon, Formal analysis, Investigation, Methodology; Martin Sachse, Methodology; Amel Bahloul, Formal analysis, Supervision, Validation, Investigation; Jean-Pierre Hardelin, Writing—review and editing; Shyam S Krishnakumar, Formal analysis, Supervision, Validation, Investigation, Visualization, Methodology; James E Rothman, Supervision, Validation; Didier Dulon, Formal analysis, Supervision, Validation, Investigation, Visualization, Methodology, Writing—review and editing; Saaid Safieddine, Conceptualization, Formal analysis, Supervision, Validation, Investigation, Visualization, Methodology, Project administration, Writing—review and editing; Christine Petit, Conceptualization, Supervision, Funding acquisition, Validation, Project administration, Writing—review and editing, Designed the study

### Author ORCIDs
Nicolas Michalski http://orcid.org/0000-0002-1287-2709
Jacques Boutet de Monvel http://orcid.org/0000-0001-6182-3527
Danica Ciric http://orcid.org/0000-0002-0098-6258
Amel Bahloul http://orcid.org/0000-0001-7042-4616
Jean-Pierre Hardelin http://orcid.org/0000-0002-1815-7909
Shyam S Krishnakumar http://orcid.org/0000-0001-6148-3251
James E Rothman http://orcid.org/0000-0001-8653-8650
Saaid Safieddine http://orcid.org/0000-0002-6159-0572
Christine Petit http://orcid.org/0000-0002-9069-002X

### Ethics
Animal experimentation: Animal experiments were carried out in accordance with European Community Council Directive 2010/63/UE under authorizations 2012-028, 2012-038, and 2014-005 from the Institut Pasteur ethics committee for animal experimentation.

Decision letter and Author response

Decision letter https://doi.org/10.7554/eLife.31013.023
Author response https://doi.org/10.7554/eLife.31013.024

## Additional files

### Supplementary files

• Transparent reporting form

DOI: https://doi.org/10.7554/eLife.31013.022

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
