## [Decision Letter]

[Editors’ note: a previous version of this study was rejected after peer review, but the authors submitted for reconsideration. The first decision letter after peer review is shown below.]

Thank you for submitting your work entitled "Otoferlin acts as a Ca^2+^ sensor for vesicle fusion and vesicle pool replenishment at auditory hair cell ribbon synapses" for consideration by *eLife*. Your article has been evaluated by a Senior Editor and three reviewers, one of whom is a member of our Board of Reviewing Editors. The reviewers have opted to remain anonymous.

Our decision has been reached after consultation between the reviewers. Based on these discussions and the individual reviews below, we regret to inform you that your work will not be considered for publication in *eLife* in its current form.

The reviewers note that this paper addresses important outstanding questions about the putative role of otoferlin as a calcium sensor for vesicle fusion in auditory hair cell ribbon synapses, an area of some controversy. While the reviewers generally found your paper impressive and liked it overall, they had substantial concerns, as you can see from their comments.

The following points will likely require more experiments:

1) What is the nature of the shift in Ca^2+^ sensitivity? Data about changes in affinity and/or cooperatively, for example using biochemistry and caged Ca^2+^ experiments combined with electrophysiology, would strengthen the authors' claims and make the observed phenotypes easier to interpret.

2) The capacitance experiments related to the endocytosis phenotype are somewhat preliminary and would benefit from more controls (Gm/Gs traces along with longer recordings post stimulation).

3) The unaltered EPSC frequency is surprising given the other phenotypes; this needs to be addressed, possibly with additional experiments. In addition, the reviewers thought there may be a need for control experiments to show that putative changes in expression levels of the mutant do not contribute to the phenotype.

*Reviewer #1:*

In this paper, the authors have developed KI mice having mutations in the Ca binding sites of Otoferin, a possible Ca sensor of exocytosis from inner hair cells. As expected, the apparent Ca sensitivity of exocytosis has been changed, and vesicle replenishment has been affected. Overall, the results are convincing, but there are several points which need authors attention.

1) It is a bit unclear how much the Ca sensitivity of mutations is altered. Some evidence from biochemistry may be helpful. Are mutations shifting the Ca sensitivity without changing the slope or has the slope (cooperativity) been changed?

2) In Figure 5, endocytotic time courses look biphasic and look a bit strange. It would be helpful to show Rs and Gm in example traces. I am wondering if fast phase of capacitance decay is due to conductance changes or not.

3) For Ca uncaging experiments, it is a pity that the authors did not perform experiments with changing intracellular Ca concentrations. As it is, it is unclear if the maximal rate of secretion has been changed (which is nothing to do with Ca sensitivity) or else if Ca sensitivity is shifted. The authors may need additional experiments or need some comments or clarification. Also, the authors should cite the works by Heidelberger/Matthews and Beutner/Moser, both of whom have developed Ca uncaging at ribbon synapses.

4) In Figure 7 (current Figure 8), I am not convinced if synaptic vesicle replenishment is Ca dependent and otoferin is the sensor. The authors need some comments.

5) Figure 8(current Figure 10) is difficult to understand. Why the EPSC frequency is not reduced in mutant mice? This is due to compensation or better conditions have to be found (higher or lower K concentrations)?

6) I do not feel that the modeling part is really required, because it is not that informative.

*Reviewer #2:*

The paper by Michalski et al. examines the molecular identity and functional properties of the calcium / release sensor at auditory hair cell ribbon synapses. To address this question, the authors combine genetic approaches (mutant mice), immunohistochemistry, electron tomography, recording of evoked potentials, presynaptic capacitance measurements, calcium imaging, calcium uncaging, and modeling. The main findings are:

- Point mutations in the C2C domain of otoferlin reduce the amplitude of auditory evoked brainstem potentials.

- Otoferlin mutations change the calcium sensitivity of exocytosis, as probed with capacitance measurements from hair cells.

- Otoferlin mutations also change release during long-lasting repetitive stimulation, suggesting a role for the protein in vesicle replenishment.

- Modeling data further suggest that the difference in the sensitivity of the calcium sensor can explain the synaptic phenotype.

Based on these findings, the authors conclude that otoferlin functions as a calcium sensor of exocytosis and vesicle replenishment in hair cell ribbon synapses. Overall, I found this a quite impressive paper. The results are interesting, because the manuscript for the first time uses the approach of Fernandez et al., 2001 (pioneered for synaptotagmin) to a different calcium-sensing protein. Furthermore, the experiments are technically well done (with some exceptions, see below), and the manuscript is concisely and accurately written. One caveat is that the role of otoferlin in exocytosis from hair cells was surrounded by controversy in the past, but I feel the present manuscript provides definitive answers to many of the outstanding questions.

1) Although otoferlin is clearly expressed in the mutant (as shown by immunocytochemistry), the exact expression levels are unknown. The authors should address this point by quantitative analysis, or at the very least put caveat sentences into the paper. If expression levels were different between wild-type and mutant, this would affect the main conclusions.

2) Is otoferlin present on synaptic vesicles, as required for a Ca^2+^ sensor mediating exocytosis? The authors might have the high-resolution imaging data to address this question.

3) The authors show that their capacitance versus calcium curves were fitted power functions with exponents less than one, suggesting the absence of cooperativity. If I read the early literature on hair cell exocytosis correctly, data were well fit by a model with five binding sites for calcium and cooperativity in the unbinding rates (b < 1; Beutner et al., 2001, Neuron). How the present data relate to these earlier findings is presently unclear. This point should be better discussed.

4) The authors generally use the heterozygous rather than wild-type mice as controls. This should be better justified in the paper. The argument that there are only minimal differences between wild-type and heterozygous animals in the auditory evoked brainstem potentials is not very compelling, because this assay may not have the sensitivity to detect small differences.

5) A design flaw in the uncaging experiments is that the absolute calcium concentration is not simultaneously monitored, as often done by the ratiometric calcium the indicator fura 2. Furthermore, the authors mention absolute calcium concentrations several times. However, it is unclear how these concentration estimates were obtained with the non-ratiometric indicator OGB5-N.

6) In the capacitance measurements, the authors only illustrate Cm traces. The authors should at least once show the corresponding gm and ga traces to convince the readers of the reliability of the measurements.

7) The endocytosis data are preliminary. First, longer traces should be shown to illustrate whether capacitance decays back to baseline. Second, the authors should analyze the decay more quantitatively, for example by exponential fitting. Finally, it is surprising that the mutant apparently does not affect endocytosis, despite its clear effects on replenishment rate. This point should be better discussed.

8) The EPSC recordings from postsynaptic boutons are a little preliminary. It is unclear how the authors determine the frequency of events and how they measure amplitude and decay time constant in complex waveforms. Finally, it is unexpected that there is no significant difference in frequency between control and mutant. Increasing the number of experiments may resolve this issue.

---

## [Author Response]

[Editors’ note: the author responses to the first round of peer review follow.]

Reviewer #1:In this paper, the authors have developed KI mice having mutations in the Ca binding sites of Otoferin, a possible Ca sensor of exocytosis from inner hair cells. As expected, the apparent Ca sensitivity of exocytosis has been changed, and vesicle replenishment has been affected. Overall, the results are convincing, but there are several points which need authors attention.1) It is a bit unclear how much the Ca sensitivity of mutations is altered. Some evidence from biochemistry may be helpful. Are mutations shifting the Ca sensitivity without changing the slope or has the slope (cooperativity) been changed?

Unfortunately, we were unable to produce and purify high‐quality C_2_C domains for biochemical assays. Three independent laboratories made several attempts with a view to probing the effects of the C_2_C domain mutations on Ca^2+^‐binding in vitro (Dr. Danica Ciric, Dr. Amel Bahloul and Sylvie Nouaille in my laboratory; Prof. R. Bryan Sutton; and Dr. Sarah Auclair and Dr. Shyam S. Krishnakumar in the laboratory of Prof. James E. Rothman).

By contrast, uncaging experiments were more successful. In the first version of the manuscript, through the various experimental protocols used to manipulate intracellular Ca^2+^ levels near release sites during Ca^2+^ channel activation, we had reported that the C_2_C domain mutations almost halved the Ca^2+^ sensitivity of the RRP sensor (i.e., decrease by a factor of 1.7 based on the modulation of depolarization levels, of 1.5 for variation of the duration of depolarization, and of 2.2 for the variation of extracellular Ca^2+^ concentrations; Figure 5). The lower Ca^2+^ sensitivity of vesicle fusion in *Otof*
^C2C/C2C^ IHCs was further established by rapid increases in intracellular Ca^2+^ concentration upon photolysis of caged Ca^2+^, which resulted in a delayed onset of exocytosis and a doubling of the time taken to reach the maximal rate of fusion (Figure 6). In the resubmitted version, we proved changes in the Ca^2+^ sensitivity of neurotransmitter release and provide a new set of Ca^2+^ uncaging data (new Figure 7). We simultaneously monitored ΔCm and intracellular Ca^2+^ concentration in inner hair cells (IHCs) loaded with the low‐affinity fluorescent Ca^2+^ indicator, OGB‐5N. This approach made it possible to show ex vivo that the Ca^2+^ sensitivity of release is reduced in *Otof*
^C2C/C2C^IHCs. Our results indicate that C_2_C domain mutations strongly decrease the Ca^2+^ affinity of otoferlin, by a factor of about 4, but not the cooperativity of release.

Biochemistry: We attempted to produce and purify wild‐type and Ca^2+^ binding mutant (D519A/D521A) versions of the otoferlin C_2_C domain for biochemical validation of the importance of the Ca^2+^‐binding site in the otoferlin C_2_C domain. The chosen domain boundaries of otoferlin C_2_C were based on the known structural properties of C_2_ domains: prediction of eight β‐strands, the periodicity of alternating hydrophobic residues in betastrands consistent with known C_2_ domain structures, identification of betabulge residues to ascertain C_2_ domain topology, geometric arrangement of consensus Ca^2+^ binding residues, location of β‐strand 3 with consensus residues conserved in all C_2_ domains, and overall length of the domain. Once these criteria were met, a 3D model was generated with Modeller, by aligning the selected primary sequence with a template appropriate for C_2_ domain topology. For confirmation that the hydrophobic core and β‐sheet assignments of the model domain were stable, the model was simulated for 100 ns with NAMD. This led us to test two different C2C domain boundaries (residues 403‐554 and residues 449‐600). We used a bacterial system to test various expression and growth parameters, the two different protein boundaries (residues 403‐554, Clone #1 and residues 449‐600, Clone #2), additives and affinity tags (poly‐histidine, GST and SUMO). Despite this comprehensive analysis, most of the clones resulted in little or no protein production. The only exception was Clone #1, with the C‐terminal His tag. However, the proteins produced from this clone were of very poor quality, giving rise to multiple bands on protein gels. This was confirmed by mass spectrometry and Edman N‐terminal sequencing analysis, which also showed additional peaks and sequence variability, respectively. Attempts to purify the otoferlin C_2_C domain further, by removing contaminants through size‐exclusion and affinity chromatography, were unsuccessful. We nevertheless performed the liposome aggregation, membrane binding, and liposome fusion assays described in a previous study (Johnson and Chapman, 2010), to check the effect of Ca^2+^‐binding, but we observed no effect of Ca^2+^ addition on the WT or mutated C_2_C domains, probably because of the suboptimal quality of the proteins used.

**Author response image 1. respfig1:** SDS-PAGE analysis showing the poor quality of the Otoferlin C_2_C proteins purified.

2) In Figure 5, endocytotic time courses look biphasic and look a bit strange. It would be helpful to show Rs and Gm in example traces. I am wondering if fast phase of capacitance decay is due to conductance changes or not.

In IHCs, two modes of endocytosis with different kinetics have been described: a slow phase with a linear C_m_ decline (in the range of 2‐10 fF/s) upon short depolarizations (typically < 50 ms in our recording conditions), and a fast phase with an exponential decline (with a time constant in the 2‐4 s range) upon longer depolarizations (typically > 100 ms) (Moser and Beutner, 2000, Beutner et al., 2001, Neef et al., 2014). In the former and current Figure 5, the fast Cm decline (in the range of 10 ms) corresponds to the transient change in conductance following depolarization and not endocytosis. We systematically record C_m_, G_m_, and G_s_ traces shortly after the end of the depolarization, to make sure that these transients are not included in our ΔC_m_ estimate. As suggested, to avoid confusion, we now provide example C_m_, G_m_, and G_s_ traces and illustrate how we evaluate ΔC_m_ (see Figure 5—figure supplement 1). We have also expanded the time scale in the new Figure 5, so that this fast decay phase can clearly be seen to result from a change in conductance.

3) For Ca uncaging experiments, it is a pity that the authors did not perform experiments with changing intracellular Ca concentrations. As it is, it is unclear if the maximal rate of secretion has been changed (which is nothing to do with Ca sensitivity) or else if Ca sensitivity is shifted. The authors may need additional experiments or need some comments or clarification. Also, the authors should cite the works by Heidelberger/Matthews and Beutner/Moser, both of whom have developed Ca uncaging at ribbon synapses.

As mentioned above (see response to point 1), we now provide a new set of Ca^2+^ uncaging experiments (new Figure 7), in which we simultaneously monitored ΔCm and intracellular Ca^2+^ concentration in IHCs loaded with OGB‐5N. Our results indicate that C_2_C domain mutations strongly reduce Ca^2+^ affinity (by a factor of about four), without changing apparent cooperativity.

Heidelberger/Matthews and Beutner/Moser are now cited at the start of the Ca^2+^ uncaging Results section.

4) In Figure 7 (current Figure 8), I am not convinced if synaptic vesicle replenishment is Ca dependent and otoferin is the sensor. The authors need some comments.

Synaptic vesicle replenishment has been shown to be Ca^2+^‐dependent and we refer to the original article (Schnee et al., 2011) in the manuscript. The grounds for our conclusion concerning the role of otoferlin as a Ca^2+^ sensor in synaptic vesicle replenishment are as follows:

i) Endocytosis is normal in *Otof*
^C2C/C2C^ IHCs. In accordance with the reviewers’ suggestions, we now provide a more detailed description of endocytosis (see response to point 2). The fast and slow modes of membrane retrieval were similar in *Otof*
^C2C/+^ and *Otof*
^C2C/C2C^ IHCs. Accordingly, vesicle pool numbers at the ribbon synapses were also normal. Moreover, rapid and global increases in intracellular Ca^2+^concentration in response to UV‐flash photolysis led to similar total synaptic release levels in *Otof*
^C2C/+^ and *Otof*
^C2C/C2C^ IHCs, strongly suggesting that endocytosis and vesicle reformation are not affected in *Otof*
^C2C/C2C^ IHCs. This particular point is now discussed as follows: “Finally, the RRP replenishment defect in *Otof*
^C2C/C2C^ IHCs was partially rescued, in terms of total synaptic release, by making Ca^2+^ available at high concentrations throughout the cytoplasm, as in Ca^2+^ uncaging experiments, suggesting that synaptic vesicle reformation is not affected in *Otof*
^C2C/C2C^ IHCs.” Moreover, this result indicates that the loss of Ca^2+^ sensitivity does not prevent the total synaptic pool of the IHCs to be depleted by high calcium loads. In previous studies of other otoferlin mutant mice, Ca^2+^ uncaging failed to elicit normal total synaptic release (Pangrsic et al., 2010, Strenzke et al., 2016).

ii) The rate of synaptic vesicle replenishment was lower in *Otof*
^C2C/C2C^ IHCs than in *Otof*
^C2C/+^ IHCs, as shown by long depolarizations or trains of 50 successive short depolarizations, suggesting an effect of C_2_C mutations on Ca^2+^‐dependent vesicle replenishment. Moreover, paired pulse ratio experiments, which probe synaptic vesicle pool replenishment efficiency independently of fusion efficiency, showed that peak vesicle pool replenishment efficiency in *Otof*
^C2C/C2C^ IHCs was only half that in *Otof*
^C2C/+^ IHCs, excluding the possibility that the synaptic vesicle pool replenishment defect was merely a consequence of the impairment of RRP fusion.

iii) In *Otof*
^C2C/C2C^ IHCs with low intracellular Ca^2+^‐buffering conditions, the exocytotic response was rapidly saturated for depolarizations lasting 15‐20 ms (Figure 5B2). This saturation is apparent for depolarizations of up to 100 ms (only data up to 50 ms are shown in Figure 5B2). In *Otof*
^C2C/C2C^ IHCs, release begins to increase again for depolarizations lasting more than 200 ms (see Figure 8). This result suggests that vesicle fusion and synaptic vesicle replenishment can be at least partially uncoupled in *Otof*
^C2C/C2C^ IHCs. This is described only briefly in the Results section, because this description precedes the synaptic vesicle replenishment section: “Remarkably, unlike the exocytotic response of *Otof*
^C2C/+^ IHCs, which did not plateau for depolarizations lasting up to 50 ms, that of *Otof*
^C2C/C2C^ IHCs rapidly saturated for depolarizations lasting 15‐20 ms (Figure 5B2), and was insensitive to 5 mM intracellular EGTA, suggesting that vesicle pool replenishment at the release sites was also impaired in these cells (Figure 5B2).” This result provides evidence that the replenishment deficit is not only due to impaired synaptic fusion, but also a direct consequence of the C_2_C domain mutation.

Based on all these findings, we conclude that otoferlin functions as a Ca^2+^ sensor for vesicle replenishment.

5) Figure 8 (current Figure 10) is difficult to understand. Why the EPSC frequency is not reduced in mutant mice? This is due to compensation or better conditions have to be found (higher or lower K concentrations)?

A number of important factors contributed to the result presented in the initial version of the manuscript. Firstly, due to the difficulty obtaining reliable bouton recordings, our first submission included a limited dataset. We now present a total of *n* = 8 recordings for each genotype. Secondly, in the previous version of the paper, we calculated EPSC activation frequency throughout the length of each recording, regardless of its length. This led to a non‐uniform comparison of EPSC activation frequencies. Finally, the application of high levels of K^+^ typically induces a rapid increase in activation frequency within a few seconds, followed by a slowing over tens of seconds or minutes. This first phase is greatly influenced by IHC release capacity. For this reason, we now evaluate EPSC activation frequency in the first 10 seconds after the application of 25mM K^+^. It is now clear that the mutation has a strong effect on EPSC rate (see new Figure 10). The results and Discussion sections have been modified accordingly.

6) I do not feel that the modeling part is really required, because it is not that informative.

We think that the first part of the modeling results is noteworthy (new Figure 11) because these results show, for the first time, that this ‘simple’ model of vesicle release can account for the superlinear phase of release (Schnee et al., 2005, Schnee et al., 2011). In addition, the model shows that our experimental results can be mimicked only by a decrease in RRP fusion rate (*K*_1_) and the rate of transition from the recycling pool to the RRP (*K_2_*), but that there is no change in Ca^2+^ thresholds for the recruitment of each vesicle pool or in vesicle number in each pool.

We have removed the second part of the modeling results, in which the same model settings were used to reproduce our experimental results for 100 ms depolarizations.

Reviewer #2:[…] 1) Although otoferlin is clearly expressed in the mutant (as shown by immunocytochemistry), the exact expression levels are unknown. The authors should address this point by quantitative analysis, or at the very least put caveat sentences into the paper. If expression levels were different between wild-type and mutant, this would affect the main conclusions.

In the revised version, we provide a comparative analysis of otoferlin levels between *Otof*
^C2C/+^ and *Otof*
^C2C/C2C^ inner hair cells (IHCs), based on comparisons of otoferlin immunofluorescence, as previously described for other otoferlin mutants (Strenzke et al., 2016, Pangrsic et al., 2010). We found no difference at the apex, middle, or base of *Otof*
^C2C/+^ and *Otof*
^C2C/C2C^ IHCs (see new Figure 3). The antibody was also validated again in *Otof ^‐/‐^* knockout mice (see new Figure 3).

2) Is otoferlin present on synaptic vesicles, as required for a Ca^2+^ sensor mediating exocytosis? The authors might have the high-resolution imaging data to address this question.

We now provide super‐resolution images obtained by stimulated emission depletion imaging (STED) microscopy, of IHCs stained for otoferlin and a marker of IHC vesicles (Vglut3) (see new Figure 3—figure supplement 2). Using nearest neighbor analysis, we were able to show that the distance between a randomly chosen otoferlin spot and the nearest Vglut3 spot (quantifying the colocalization of the two proteins) was similar in *Otof*
^C2C/+^ and *Otof*
^C2C/C2C^ IHCs, and significantly smaller than the mean distance between a random Vglut3 spot and its nearest neighboring Vglut3 spot (the expected distribution of nearest‐neighbor distances for randomly distributed points). The distributions of distances between a given otoferlin immunostaining spot and the closest Vglut3 immunostaining spot were also similar in *Otof*
^C2C/+^ and *Otof*
^C2C/C2C^ IHCs, suggesting that the C_2_C mutations did not affect the association of otoferlin with synaptic vesicles.

3) The authors show that their capacitance versus calcium curves were fitted power functions with exponents less than one, suggesting the absence of cooperativity. If I read the early literature on hair cell exocytosis correctly, data were well fit by a model with five binding sites for calcium and cooperativity in the unbinding rates (b < 1; Beutner et al., 2001, Neuron). How the present data relate to these earlier findings is presently unclear. This point should be better discussed.

These two sets of experiments cannot be directly compared because of their very different experimental conditions. In the case of ΔC_m_ measurements in response to IHC depolarization (Figure 5), Ca^2+^ enters the IHC locally via the Ca^2+^ channels. In addition, in the recording conditions of Figure 5, the recruited vesicles were mostly those of the RRP (a total release of 20fF). By contrast, in Ca^2+^uncaging experiments, Ca^2+^ is uncaged close to the plasma membrane and within the cytoplasm simultaneously, and almost instantaneously. Both RRP vesicles and vesicles from the reserve pools are massively recruited (a total of 1000 to 2000 fF).

For ΔC_m_ measurements in response to IHC depolarization (Figure 5), we obtained a linear relationship in *mature* control IHCs, as described in many other studies (see, for example, (Johnson et al., 2010, Johnson et al., 2005)). [It should be noted that this relationship is not linear in *immature* IHCs]. In this experimental setting, we plotted ΔC_m_ as a function of I_Ca_ at the end of the stimulus, so we had no access to the early kinetics of release during depolarization. In the Ca^2+^ uncaging experiments reported by Beutner et al. (Beutner et al., 2001), the initial release rate (dCm/dt) is plotted. In these uncaging experiments, if the final ΔC_m_ were plotted as a function of I_Ca_, a flat relationship would probably have been observed, as suggested by the examples shown in figure 2 of the same paper (Beutner et al., 2001), because the same maximal C_m_ is reached whatever the original intracellular calcium concentration, suggesting that release is saturated after a certain time.

As described below (see reply to point 5), we provide new data for Ca^2+^ uncaging experiments in *Otof*
^C2C/+^ and *Otof*
^C2C/C2C^ IHCs. In control cells, we obtained cooperativity values similar to those reported by Beutner et al. (Beutner et al., 2001), between 3 and 4.

4) The authors generally use the heterozygous rather than wild-type mice as controls. This should be better justified in the paper. The argument that there are only minimal differences between wild-type and heterozygous animals in the auditory evoked brainstem potentials is not very compelling, because this assay may not have the sensitivity to detect small differences.

We now provide new ΔC_m_ recordings for *Otof*
^+/+^ IHCs, further demonstrating that synaptic release is similar in *Otof*
^C2C/+^ and *Otof*
^+/+^ IHCs. We measured Ca^2+^ currents (*I*_Ca_) and the corresponding ΔC_m_ in response to depolarizations of various amplitudes (from a holding membrane potential of ‐95 mV to potentials between ‐65 mV and +35 mV) and 20 ms duration in *Otof*
^+/+^ IHCs, and compared these results with those of *Otof*
^C2C/+^ IHCs (see Figure 5—figure supplement 1). We also subjected *Otof*
^+/+^ IHCs to periodic stimulation with a train of 50 short (5 ms long) depolarizations to ‐10 mV, separated by 10 ms intervals. We found that this protocol elicited superlinear release in around half the recorded *Otof*
^+/+^ IHCs, as for *Otof*
^C2C/+^ IHCs.

5) A design flaw in the uncaging experiments is that the absolute calcium concentration is not simultaneously monitored, as often done by the ratiometric calcium the indicator fura 2. Furthermore, the authors mention absolute calcium concentrations several times. However, it is unclear how these concentration estimates were obtained with the non-ratiometric indicator OGB5-N.

We have run additional experiments for simultaneous quantification of the increase in intracellular Ca^2+^and membrane capacitance upon UV‐flash photolysis of DM‐nitrophen for each cell (see new Figure 7). We used the Ca^2+^ dye Oregon Green Bapta‐5N (OGB‐5N) as an intracellular calcium probe because it allows continuous rapid monitoring of the Ca^2+^ signal under confocal microscopy (line scan at 1 kHz), before and after UV‐flash Ca^2+^ uncaging. OGB‐5N is a dye displaying excitation at a long wavelength (488 nm), a visible wavelength at which there is no photolysis of DM‐nitrophen, by contrast to UV epifluorescence experiments using the ratiometric dye Fura‐2 (excitation at 340 and 380 nm). Another advantage of OGB‐5N is its Ca^2+^ affinity, which is lower (*K_D_*= 20 µM) than that of Fura‐2 (*K_D_* = 140 nM), making it possible to obtain more precise measurements of variations of Ca^2+^concentration, in the μM range.

As now described in the Materials and methods section, the Ca^2+^‐dependent fluorescent signal of OGB‐5N was first calibrated in situ, in the hair cell cytoplasm, as previously described (Vincent et al., 2014). Hair cells were voltage‐clamped at ‐70 mV in the whole‐cell configuration with various solutions differing in their free calcium concentrations ([Ca^2+^]_free_), which ranged from 1 to 100 µM. The patch pipettes were filled with a Cs^+^‐based solution (145 mM CsCl, 5 mM HEPES 5, 20 mM TEA and 0.05 mM OGB‐5N) and various concentrations of Ca^2+^ and EGTA. Ca‐EGTA Maxchelator freeware (Stanford, CA, USA) was used to determine [Ca^2+^]_free_. After 2 min of cell loading and equilibration, the fluorescence signal was measured at each [Ca^2+^]_free_, in different cells. Fluorescence emission values were plotted as F/Fmax against [Ca^2+^]_free_. Data points were best fitted by a sigmoidal function with a *K_D_* of 23.30 ± 0.71 µM. This *K_D_* value is very close to the value reported in vitro (*K_D_* = 20 µM, Molecular Probes, Life Technologies). We quantified the Ca^2+^ responses during the Ca^2+^ uncaging experiments in each hair cell, by calculating intracellular [Ca^2+^] as follows ((Pawley, 2006), chapter 42): [Ca^2+^]_i_= *K_D_* (F F_min_)/(F_max_‐F) where *K_D_* is the estimated constant of dissociation, F the fluorescence at time *t*, F_min_ the minimum fluorescence and F_max_ the maximal fluorescence.

6) In the capacitance measurements, the authors only illustrate Cm traces. The authors should at least once show the corresponding gm and ga traces to convince the readers of the reliability of the measurements.

We now provide an example of C_m_, G_m_, and G_s_ traces and illustrate how we typically evaluate *Δ*C_m_(see Figure 5—figure supplement 1). We have also added another example for the long endocytosis recordings (Figure 9—figure supplement 1).

7) The endocytosis data are preliminary. First, longer traces should be shown to illustrate whether capacitance decays back to baseline. Second, the authors should analyze the decay more quantitatively, for example by exponential fitting. Finally, it is surprising that the mutant apparently does not affect endocytosis, despite its clear effects on replenishment rate. This point should be better discussed.

We now provide data for a new set of endocytosis experiments with recordings over longer durations (30 s after depolarization or until the C_m_ trace returned to baseline). We studied the two modes of endocytosis with different kinetics that have been described: the slow phase, with a linear C_m_ decline, upon short depolarizations (typically < 50 ms in our recording conditions), and the fast phase, with an exponential decline upon longer depolarizations (typically > 100 ms) (Moser and Beutner, 2000, Beutner et al., 2001, Neef et al., 2014). The kinetics of the slow and fast components of endocytosis were similar in *Otof*
^C2C/+^ and *Otof*
^C2C/C2C^ IHCs (new Figure 9). We also provide, in the supplementary data, a full example of a long capacitance recording following a short depolarization, showing the C_m_, G_m_, and G_s_ traces (Figure 9—figure supplement 1).

The normal endocytosis observed is consistent with the normal otoferlin levels, normal vesicle numbers at ribbon synapses, and the similar total synaptic release in *Otof*
^C2C/+^ and *Otof*
^C2C/C2C^ IHCs upon rapid and global increases in intracellular Ca^2+^ upon Ca^2+^ uncaging. Together, these results strongly suggest that endocytosis and vesicle reformation are not affected in *Otof*
^C2C/C2C^ IHCs, contrary to reports for other mouse otoferlin mutants (Pangrsic et al., 2010, Strenzke et al., 2016). Vesicle reformation is now discussed as follows: “Finally, the RRP replenishment defect in *Otof*
^C2C/C2C^ IHCs was partially rescued, in terms of total synaptic release, by making Ca^2+^ available at high concentrations throughout the cytoplasm, as in Ca^2+^ uncaging experiments, suggesting that synaptic vesicle reformation is not affected in *Otof*
^C2C/C2C^ IHCs.”

8) The EPSC recordings from postsynaptic boutons are a little preliminary. It is unclear how the authors determine the frequency of events and how they measure amplitude and decay time constant in complex waveforms. Finally, it is unexpected that there is no significant difference in frequency between control and mutant. Increasing the number of experiments may resolve this issue.

A number of important factors contributed to the result presented in the initial version of the manuscript. Firstly, due to the difficulty obtaining reliable bouton recordings, our first submission included a limited dataset. We now present a total of *n* = 8 recordings for each genotype. Secondly, in the previous version of the paper, we calculated EPSC activation frequency throughout the length of each recording, regardless of its length. This led to a non‐uniform comparison of EPSC activation frequencies. Finally, the application of high levels of K^+^ typically induces a rapid increase in activation frequency within a few seconds, followed by a slowing over tens of seconds or minutes. This first phase is greatly influenced by IHC release capacity. For this reason, we now evaluate EPSC activation frequency in the first 10 seconds after the application of large amounts of K^+^. It is now clear that the mutation has a strong effect on EPSC rate (see new Figure 10). The Results and Discussion sections have been modified accordingly.

The EPSC methods are now detailed as follows: “EPSC recordings were performed with a MultiClamp 700B amplifier (Molecular Devices), and a National Instruments digitizer (NI‐PCIe 6351). […] Decay time constants were calculated with monophasic recorded EPSCs only.”